# Human consumption of seaweed and freshwater aquatic plants in ancient Europe

Stephen Buckley [1] ✉, Karen Hardy [2] ✉, Fredrik Hallgren [3], Lucy Kubiak-Martens[4], Žydrūnė Miliauskienė[5], Alison Sheridan [6], Iwona Sobkowiak-Tabaka[7] & Maria Eulalia Subirà [8]

During the Mesolithic in Europe, there is widespread evidence for an increase in exploitation of aquatic resources. In contrast, the subsequent Neolithic is characterised by the spread of farming, land ownership, and full sedentism, which lead to the perception of marine resources subsequently representing marginal or famine food or being abandoned altogether even at the furthermost coastal limits of Europe. Here, we examine biomarkers extracted from human dental calculus, using sequential thermal desorption- and pyrolysis-GCMS, to report direct evidence for widespread consumption of seaweed and submerged aquatic and freshwater plants across Europe. Notably, evidence of consumption of these resources extends through the Neolithic transition to farming and into the Early Middle Ages, suggesting that these resources, now rarely eaten in Europe, only became marginal much more recently. Understanding ancient foodstuffs is crucial to reconstructing the past, while a better knowledge of local, forgotten resources is likewise important today.

Seaweeds comprise approximately 10,000 species of macroalgae that live in intertidal and subtidal zones to around 300 m deep around many of the world's coastlines. Around 145 species of seaweed are eaten today, principally in Asia[1], sometimes with considerable health benefits[2]. Archaeological evidence for seaweed is only rarely recorded[3–5] and is almost always considered in terms of non-edible uses[6–9], although seaweed was detected in masticated cuds from the 12,000-year-old site of Monte Verde, Chile[3]. Related molluscs suggest the presence of seaweed on some archaeological sites;[4,5] however, direct evidence for their human consumption in the past has been lacking and they have therefore not been considered as part of ancient European diet. Their presence on archaeological sites has been related to non-edible uses, including fuel[6–8], food wrappings[9,10], fertiliser[11] and cramp (vitrified seaweed and sand) linked to gathering bone fragments during cremation[12]. A text attributed to St Columba (521–597 AD)

recounts collecting dulse (red seaweed) in Scotland[13]. Historical accounts report laws related to collection of seaweed in Iceland[14], Brittany[15] and Ireland[16] dating to the 10th century AD and the broad use of seaweed as animal fodder in northwest Europe[17]. By the 18th century seaweed was considered as famine food even in the Scottish islands[18], although some coastal regions still consume seaweed today and laverbread (*Porphyra umbilicalis*) is still eaten in Wales[19]. Freshwater aquatic plants (macrophytes) continue to be economically important in parts of Asia, nutritionally[20] and medicinally[21]. Although archaeological evidence for roots and tubers of the freshwater aquatic plants yellow water lily (*Nuphar lutea*), white water lily (*Nymphaea alba*) and pondweed (*Potamogeton pectinatus*) is present on Mesolithic[22–24] and Neolithic[25–27] sites across Europe, they have not generally been considered as part of the ancient European diet. Sea kale (*Crambe maritima*) is mentioned by Pliny as a sailor's anti-scurvy remedy[28] and by

[1]Department of Archaeology, University of York, Kings Manor, Exhibition Square, York YO1 7EP, UK. [2]Department of Archaeology, University of Glasgow, Molema Building, Lilybank Gardens, Glasgow G12 8RZ, UK. [3]The Cultural Heritage Foundation, Stiftelsen Kulturmiljövård, Pilgatan 8D, 721 30 Västerås, Sweden. [4]BIAX Consult, Symon Spiersweg 7 D2, 1506 RZ Zaandam, The Netherlands. [5]Department of Anatomy, Histology and Anthropology, Institute of Biomedical Sciences, Faculty of Medicine, Vilnius University, Vilnius, Lithuania. [6]Department of Scottish History and Archaeology, National Museums Scotland, Chambers Street, Edinburgh EH1 1JF, UK. [7]Faculty of Archaeology, Adam Mickiewicz University in Poznań, Uniwersytetu Poznańskiego 7, 61-614 Poznań, Poland. [8]GREAB, Unitat d'Antropologia Biològica, Departament de Biologia Animal, Biologia Vegetal i Ecologia. Facultat de Biociències. Universitat Autònoma de Barcelona, Barcelona, Spain. ✉e-mail: sb55@york.ac.uk; karhardy2@gmail.com

Mrs Beeton as a 'type of asparagus', following botanist William Curtis's recommendation of its use as a vegetable in Britain[29]. It is known as a vitamin C-rich vegetable, despite its use falling in recent times due to over-exploitation[30].

The earliest evidence for cereal crop domestication in southwest Asia dates to around 13,000 years ago[31]. The Neolithic period, which began around 10,500 years ago, became widespread initially in southwest Asia[32] then gradually spread through Europe. It was well established in southern Iberia by around 7500 years ago[33] and the far north of Scotland around 6000 years ago. Although the use of grinding tools, management of resources and storage of surplus had all existed in some places before the Neolithic, once these came together with the habitual use of pottery and an economy that was based predominantly on fully domesticated animals and plants, this ultimately established the social and economic foundations that underpin today's world, including the full control and management of terrestrial food sources, land ownership, population increase and full sedentism. The Mesolithic period that immediately preceded this was based on exploitation of wild resources and is particularly characterised by aquatic resource exploitation[34]. The perceived switch to terrestrial resources at the start of the Neolithic has driven the perception of marine resources as subsequently representing marginal or famine food[35,36], leading to the suggestion that they were abandoned altogether in favour of animal and, notably, dairy produce in the Neolithic[37], even at the furthermost coastal limits of Europe[38].

Here, we report the first direct evidence in the form of identifiable and characteristic biological marker ('biomarker') compounds extracted from samples of human dental calculus for widespread consumption of coastal resources, including seaweed, submerged aquatic plants (macrophytes) and in one location *Crambe maritima* (sea kale) from across Europe covering the period from the Mesolithic through the adoption of agriculture and later prehistoric periods up to the Early Middle Ages. This suggests that these resources, rarely eaten in Europe today, were, until relatively recently, an habitual part of the diet supporting the historical evidence that suggests that they only latterly became marginal or famine resources and animal fodder. Understanding how the use of food resources has altered over time is crucial to reconstructing the past, while a better knowledge of early human diet and forgotten local resources, can provide clues to assist with improving today's diet and the environmental impact of food supply. Recovery and identification of dietary components from characteristic biomarkers embedded in dental calculus is a unique way to obtain direct evidence of identifiable ingested items in archaeological populations (Fig. 1). A biomarker in this context is an organic compound that is characteristic of the original molecule and can survive over archaeological and geological time periods in a structurally recognisable form, allowing it to be correlated with the source material. Dental calculus is common on skeletal remains from most archaeological periods and acts as a store for biomolecules that have been ingested during life[39].

## Results

Dental calculus samples from 74 individuals from 28 archaeological sites across Europe from north Scotland to southern Spain (Fig. 2) were investigated using sequential thermal desorption-gas chromatography-mass spectrometry (TD-GC-MS) and pyrolysis-gas chromatography-mass spectrometry (Py-GC-MS) (Supplementary Data 2–5). This facilitates the identification of both free/unbound and polymerised/bound organic components[40], which can be present in many archaeological organic residues including the organic component in dental calculus. Of these, 37 samples (33 individuals) had identifiable chemical biomarkers (Supplementary Table 1/Supplementary Data 1) indicating consumption of fats/oils, proteins, carbohydrates, and evidence for exposure to fire/cooking (Supplementary Information) and in one case, a biodegraded/archaeological beeswax or propolis wax (Table 1) supporting previous evidence for its use in prehistory[41] (Supplementary Notes). However, characteristic biomarkers that identify ingestion of aquatic resources were also present. Specifically, seaweed (macroalgae), freshwater algae and aquatic plants were also identified in 26 samples on the basis of their distinct, unusual and complex organic chemistry, with each providing its own diagnostic suite of highly resilient biomarkers from three different compound classes: alkylpyrroles, amino acids and lipids.

Recovery and identification of dietary components from biomolecular markers embedded in dental calculus is a unique way to obtain direct evidence of ingested items in archaeological populations[39]. The organic component within the dental calculus can form a biopolymeric component—'chemical fossil'. This makes it intractable to conventional organic residue analysis yet the biomarkers that constitute the monomeric building blocks of the organic bound/polymeric component can be characterised and identified by Py-GC-MS[40].

### Seaweed

As a 'chemical fossil', algae can be expected to reveal a distinctive suite of alkylpyrroles when analysed by pyrolysis-GC-MS[42]. Key research in this area studied the composition of kerogen, i.e., macromolecular fossil organic matter, from the Miocene Monterey Formation, California, USA[42] that contains a significant number of large soft-bodied seaweeds, which are rarely found as fossils elsewhere[43], and based on geological and organic geochemical data algae are understood to be the primary source of organic matter[44] (see Supplementary Information pp. 57–58). The same suite of alkylpyrroles identified in the Monterey kerogen were also observed in the pyrogram of thirteen samples representing thirteen individuals from Mesolithic Casa Corona, Spain, the Neolithic chambered cairns of Isbister and Quanterness, Orkney, and from Distillery Cave, Scotland (Supplementary Information). These $C_1$ to $C_6$ alkyl pyrroles are highly diagnostic of the presence of a significant tetrapyrrole component and their porphyrin-derived origin, combined with the relative abundance of some of the key biomarkers, has been used to identify potential algal sources in fossils[42,44] and would be expected in similarly diagenetically altered archaeological material.

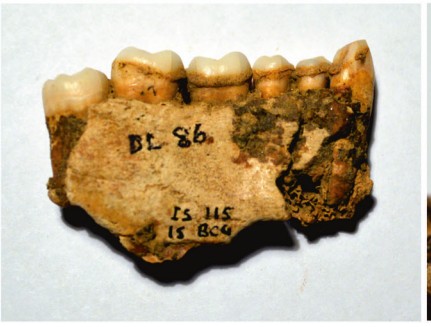 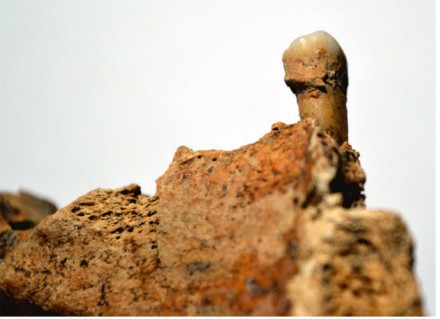

**Fig. 1 | Examples of dental calculus extracted and analysed.** Both samples are from Isbister, Orkney. Left is sample DL 188.1, right is sample DL 86.

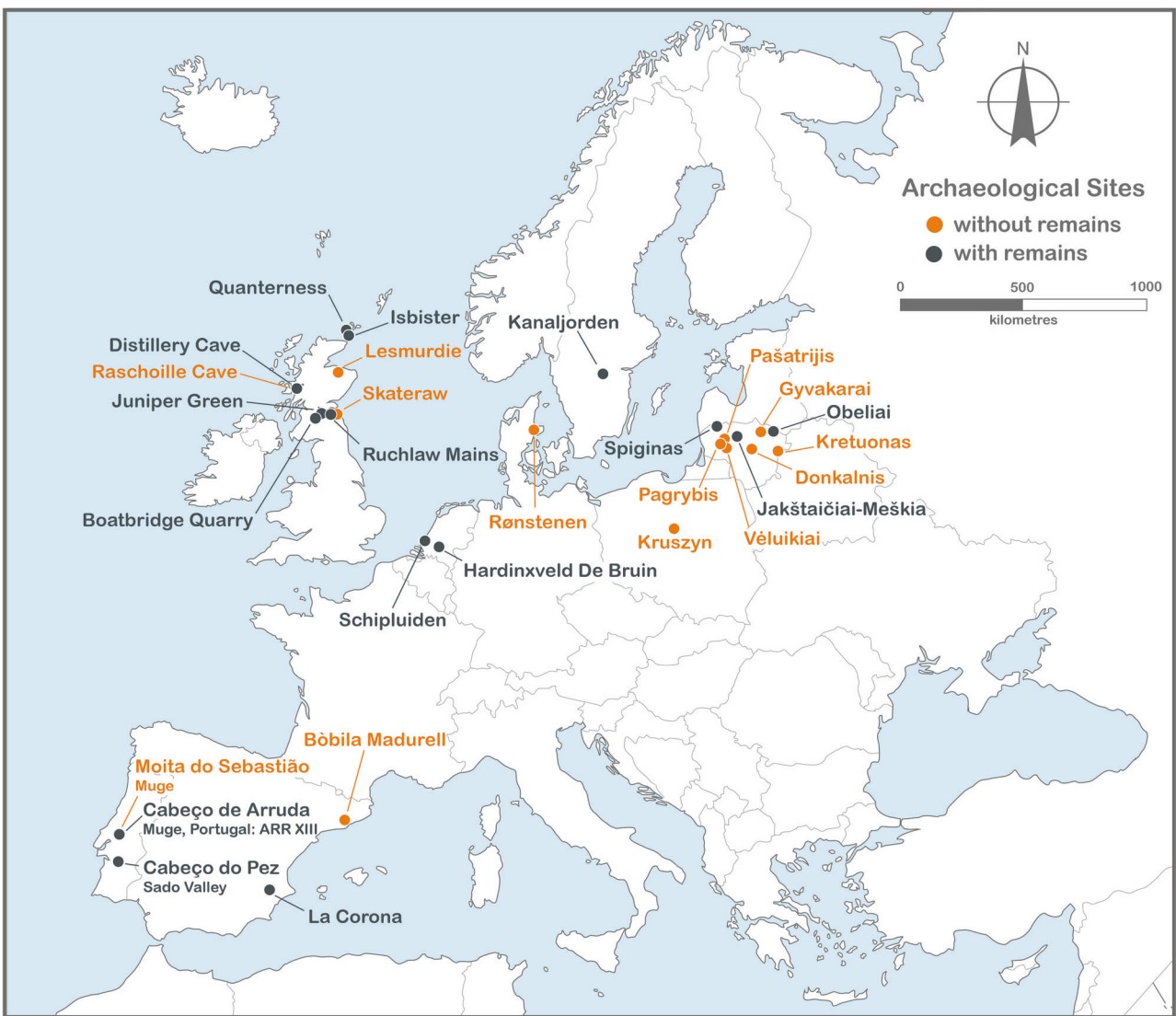

**Fig. 2 | Studied archaeological sites.** Samples from twenty-eight sites were analysed. Of these, samples from 15 sites produced identifiable biomarkers extracted from samples of dental calculus (Image produced using Corel Draw V23).

Sample DL5838 (SK77), Isbister, which displayed the greatest degree of biomolecular preservation, revealed essentially the same alkyl pyrrole profile as putative algal-derived kerogen[42,44] and all six samples with $C_1$ to $C_{5+}$ alkylpyrroles contained the same alkyl pyrroles consistent with a macroalgae, i.e., seaweed (Fig. 3, Table 2, Supplementary Data 2, 3 and 4). More specifically and diagnostically, unusual alkyl pyrroles derived from the $C_8$ side chain of bacteriochlorophylls c and d[44] were detected in five of these samples and included 3-methyl-4-iso-butylpyrrole, 2,3-dimethyl-4-*n*-propylpyrrole, 3-ethyl-4-isobutylpyrrole, 3-methyl-4-neopentylpyrrole and 2-methyl-3-ethyl-4-*n*-propylpyrrole (see Table 2). These specific alkyl pyrroles are biomarkers for the relatively rare bacteriochlorophylls c and d which are found in the green bacteria phylum *Chloroflexi* (e.g., *Chloroflexus* sp.)[45]. *Chloroflexi* are known to naturally occur in moderate abundance in association with intertidal and subtidal red seaweeds such as *Porphyra umbilicalis*[46,47]. Notably, an extensive review of 161 studies on bacteria associated with green, brown and red seaweeds found *Chloroflexi* only in association with red seaweed[48].

Although diagenetic changes may have altered the original protein composition, protein markers observed in the Py-GC-MS of these same samples, also indicate high levels of the aromatic amino acids phenylalanine, tyrosine and tryptophan, in addition to significant

levels of glycine, alanine, cysteine, proline (and possibly hydroxypro-line), serine, arginine, glutamic acid and aspartic acid (see Supplementary Information). Seaweeds (macroalgae), especially red types, can contain up to 30–50% protein (dry weight)[49,50] (Supplementary Information p.62) while green, brown and red types contain significant amounts of these particular amino acids[49,50] (Supplementary Information p.62), with some species being higher in phenylalanine, tyrosine and tryptophan than egg proteins[50].

The fatty acids constituting the original acyl lipids were not observed in the TD-GC-MS, confirming an absence of these free lipid biomolecules; however, the alkene/alkanes dominant in the pyrogram are indicative of a biopolymer deriving from these labile acyl lipids, following oxidative cross-linking of unsaturated and saturated fatty acids[51,52]. The bimodal distribution of *n*-1-alkenes/*n*-alkanes with maxima at $C_{11}/C_{12}$ and $C_{14}$, combined with abundant short chain ($C_4$-$C_7$) *n*-1-alkenes, suggests the original acyl lipids are likely to have been high in polyunsaturated fatty acids, consistent with a seaweed source. Green, brown and red seaweeds differ in the relative abundances of specific unsaturated fatty acids present in their lipid component[1,49,53–55] (see also Supplementary Information pp.66–68).

Green seaweeds are high in $C_{16}$ and $C_{18}$ polyunsaturated fatty acids, with $C_{20}$ acids in minor abundance and $C_{22}$ acids very minor or

## Table 1 | All samples containing recovered chemical biomarkers of seaweed, freshwater aquatic plants, freshwater micro/macroalgae and beeswax

| Site | Sample size,mg | Sex/ age | Date/ period | Results |
|---|---|---|---|---|
| Spiginas S A5 Lithuania | 1.63 | F, 30–40 | Mesolithic 6412–6258 cal BC | Submerged freshwater aquatic plant (Nymphaea spp.) |
| La Corona, Spain: 25 | 5.68 | F, 35–40 | Mesolithic 6059–5849 cal BC | Seaweed. Meat/fish |
| Cabeço do Pez, Portugal: b251-b | 1.77 | F | Mesolithic 5850–5610 cal BC | Submerged freshwater aquatic plant (Potamogeton spp. pondweed) |
| Cabeço do Pez, Portugal: Esq27 Esquina | 4.06 | M | Mesolithic 5850–5610 cal BC | Submerged freshwater aquatic plant (Potamogeton spp. pondweed) |
| Cabeço do Pez, Portugal: 6250-D Esquina A-B Jordi | 2.65 | M | Mesolithic 5850–5610 cal BC | Submerged freshwater aquatic plant (Potamogeton spp. pondweed) |
| Cabeço da Arruda, Portugal: ARR XIII (113) | 5.70 | F | Mesolithic | Beeswax |
| Distillery Cave, Oban, Scotland. Adult male "1': Canine right lingual | 5.62 | M | Early Neolithic c. 3700 BC | Seaweed (red) |
| Distillery Cave, Oban, Scotland. Box 6, ·3': PM2 right labial | 4.58 | M (prob.) | Early Neolithic c. 3700 BC | Seaweed |
| Quanterness, Orkney Box 10/Box K/bag 203 (Bag 3) 249-01: M2 right lingual | 5.38 | | Middle to Late Neolithic c. 3200–2800 BC | Seaweed (red) |
| Quanterness, Orkney Box 20/Bag 675 (Box A: Bag 2) 2552-02: M2 right lingual | 9.31 | | Middle to Late Neolithic c. 3200–2800 BC | Seaweed (red) |
| Isbister, Orkney, Scotland DL42 Sk8 SC1 ST2 | 2.57 | M 40–50 | Middle to Late Neolithic 3200–2800 BC | Seaweed. Meat/fish/dairy Possible exposure to fire/cooking |
| Isbister Orkney, Scotland DL2I04 Sk11 BC8/31 SC3 IS1976 | 3.19 | M 17–23 | Middle to Late Neolithic 3200–2800 BC | Seaweed. Meat/fish/dairy Possible exposure to fire/cooking |
| Isbister Orkney, Scotland DL86 Sk78 BC4 IS115 mandible IS1976 | 4.89 | | Middle to Late Neolithic 3200–2800 BC | Seaweed. Meat/fish/dairy |
| Isbister, Orkney, Scotland DL170 Sk78 SC1 ST2 mandible IS1958 | 3.74 | | Middle to Late Neolithic 3200–2800 BC | Seaweed, Meat/fish/dairy |
| Isbister, Orkney, Scotland DL715 24 Sk77 BC4 mandible IS1976 | 12.15 | | Middle to Late Neolithic 3200–2800 BC | Seaweed (red) Meat/fish/dairy |
| Isbister, Orkney, Scotland DL5083 x3 Sk77 BC2c IS128 mandible IS1976 | 10. 39 | | Middle to Late Neolithic 3200–2800 BC | Seaweed Meat/fish/dairy |
| Isbister, Orkney, Scotland DL5119 x4 Sk77 BC6(3) ST5 mandible 2 IS1976 | 6.03 | | Middle to Late Neolithic 3200–2800 BC | Seaweed Meat/fish/dairy |
| Isbister, Orkney, Scotland DL5838 Sk77 BC5 IS113 incisor IS1976 | 10.95 | | Middle to Late Neolithic 3200–2800 BC | Seaweed (red) Meat/fish/dairy. Leafy green vegetables Possible exposure to fire/cooking |
| Boatbridge Quarry, Thankerton, Scotland cist 2, (Beaker NMS X.EG 106): PM2 left labial S1 | 2.60 | | Chalcolithic/Early Bronze Age 2460–2140 cal BC | Submerged freshwater aquatic plant (Nymphaea spp.) Exposure to fire/cooking |
| Boatbridge Quarry, Thankerton, Scotland cist 2, (Beaker NMS X.EG 106): M2 right labial S2 | 1.33 | | Chalcolithic/Early Bronze Age 2460–2140 cal BC | Submerged freshwater aquatic plant (Nymphaea spp.) Exposure to fire/cooking |
| Juniper Green, Central Scotland(NMS X.ET 33): M3 | 2.48 | | Chalcolithic/Early Bronze Age 2335–2135 cal BC | Submerged freshwater aquatic plant; (Nymphaea spp.) |
| Obeliai, Lithuania Sk1128, T32 labial, 1A5 | 6.83 | M 50–55 | C5–6 AD | Submerged freshwater aquatic plant (Nymphaea spp.) Exposure to fire/cooking |
| Obeliai. Lithuania Sk1289. T32 labial, 1A8 | 5.09 | M 30–35 | C5–6 AD | Freshwater micro/macroalgae |
| Obeliai, Lithuania Sk1294 T43 lingual, 1A9 | 13.87 | M 50–55 | C5–6 AD | Freshwater micro/macroalgae Meat/fish/dairy |
| Obeliai, Lithuania Sk0984. T33 lingual, 1A10 | 3.74 | F > 55 | C5–6 AD | Submerged freshwater aquatic plant (Nymphaea spp.) |
| Jakštaičiai-Meškia, Lithuania Sk1386 T46 lingual, 1A32 | 8.47 | F 20–25 | C7–12 AD | Freshwater micro/macroalgae |
| Jakštaičiai-Meškia, Lithuania Sk1386 T36 lingual, 1A33 | 11.05 | M 35–40 | C7–12 AD | Freshwater micro/macroalgae |

See Supplementary Information for all other results, description of the sites and detailed discussions of these findings.

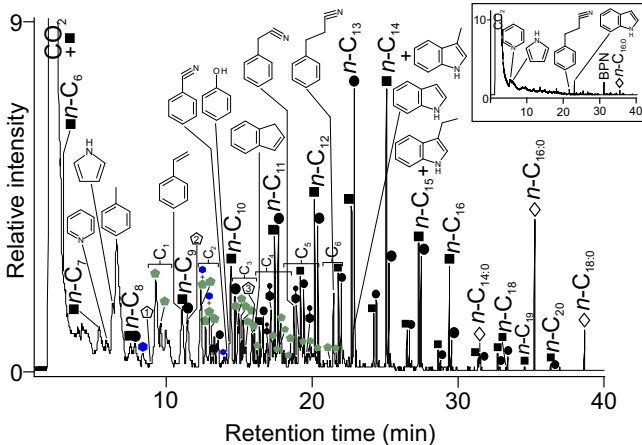

**Fig. 3 | Reconstructed total ion chromatogram of the pyrogram (pyrolysis profile) (610 °C for 10 s) of sample DL5838, after thermal desorption (310 °C for 10 s).** Key: Peak identities (x indicates carbon chain length): filled squares, Cx indicates alkenes; filled circles, Cx indicates alkanes; open diamonds, Cx:y indicates acyclic nitriles; blue filled hexagons indicates alkyl pyridines; green filled pentagons indicates alkyl pyrroles (with alkyl chain length above, Cx); filled hexagons with filled circles attached indicates alkyl phenols; open pentagons with numerals inside indicates carbohydrate pyrolysis markers, 1 is 2-methylfuran, 2 is 2-methyl-2-cyclopenten−1-one and 3 is 2,3-dimethyl-2-cyclopentene−1-one. Also shown are the structures of twelve aromatic compounds identified: pyridine, pyrrole, toluene, styrene, benzonitrile, phenol, indene, benzyl nitrile, benzenepropanenitrile, indole, 3-methylindole (skatole) and 3-ethylindole. $CO_2$ indicates carbon dioxide. Inset displays a reconstructed total ion chromatogram of the thermal desorption profile (310 °C for 10 s) of this sample. Peak identities: BPN indicates a benzenepropanenitrile derivative; open diamond, Cx:y indicates an acyclic nitrile. Also shown are four aromatic compounds: pyridine, pyrrole, benezenepropanenitrile and indole. $CO_2$ indicates carbon dioxide.

absent; brown seaweeds have less abundant $C_{16}$ acids, with $C_{18}$ acids being major and $C_{20}$ acids more significant than green; red seaweeds have $C_{20}$ polyunsaturated acids dominating, with significant amounts of $C_{22}$ acids[1,49,53–55]. Although these $C_{16}$ to $C_{22}$ polyunsaturated mid- and long-chain biomolecules characteristic of a marine macroalgal input were not detected, these compounds are particularly labile and susceptible to degradation so would not have been expected to survive unaltered over archaeological time periods[56]. Notably however, the expected biopolymeric products generated during pyrolysis if these labile biomolecules were incorporated into a bound organic fraction (see also Supplementary Information) would be expected to maximise at ~$C_{11}$/$C_{12}$ and ~$C_{14}$ n-1-alkene/n-alkanes, i.e., promote the bimodal distribution of alkene/alkanes observed in these samples[40,51,52]. The pyrolysis lipid profiles observed can be compared with typical expected biomarkers profiles tentatively related to green, brown or red seaweeds. When relative abundance of the sum of n-1-alkenes/n-alkanes for carbon chain numbers $C_8$ to $C_{16}$ is plotted for green, brown and red seaweeds and for sample DL5838, Isbister, Orkney, this is most similar to red seaweeds (see Fig. 4 and Supplementary Information pp.66–68), which is consistent with the seaweed evidence revealed by the specific alkyl pyrrole biomarkers identified, most notably the rare c and d bacteriochlorophylls known to occur in association with edible intertidal red seaweeds such as *Porphyra umbilicalis*[46]. Although the $C_{12}$ and $C_{13}$ homologues are somewhat high, this may be explained either by a small input from marine fish where those with a high oil content, in particular, contain high levels of docosahexaenoic acid ($C_{22:6}$; DHA)[57] (Supplementary Information p.68), which is normally low in macroalgae, or a red seaweed high in both $C_{20:4}$ and $C_{22:6}$, such as *Palmaria palmata* (Dulse)[53], since this would therefore be expected to increase the relative abundance of the $C_{12}$ and $C_{13}$ homologues accounting for the findings observed.

## Freshwater Algae

In four samples from Lithuania (Table 1), the presence of the alkyl pyrroles $C_1$-$C_6$ were accompanied by a series of n-alkanes, ranging from $C_{15}$-$C_{17}$ to $C_{21}$/$C_{22}$ in the TD-GC-MS. The narrow range and relatively unusual mid-chain carbon numbers, maximising at $C_{17}$ or $C_{19}$, combined with an odd over even predominance, is characteristic of an algal source[58,59] while branched alkanes with carbon numbers $C_{15}$ - $C_{21}$ as minor constituents suggests some microbially-derived biodegradation. Overall, the four samples display a remarkably similar pyrolysis profile to algal-derived kerogen[42]. These results cannot distinguish between marine and freshwater algal sources, however, the proximity of freshwater lakes and the distance to the sea, over 100 km away, suggests freshwater algae. It could have been consumed as part of the diet, since it is known to have been exploited as a food source historically by the Aztecs of Mexico (from Lake Texcoco) and the Kanembu tribe of Chad (from Lake Chad)[60], although it is also possible that the entrapment of these algal biomarkers may have resulted from the drinking of water from these local lakes, particularly given the relatively low levels of these algal biomarkers in these individuals.

## Freshwater aquatic plants

Evidence for freshwater aquatic plants has been detected in samples from Portugal, Scotland and Lithuania by a distinct series of n-alkanes in the TD-GC-MS which relate to leaf/stem and rhizome waxes in these macrophytes[58,61], specifically, a biodegraded submerged aquatic plant (Table 1)[58,61]. The chemical profile for all three samples from Cabeço do Pez, Portugal ($C_{19}$ to $C_{29}$ n-alkanes, maximising at $C_{23}$), is consistent with a submerged aquatic plant such as *Potamogeton* spp. (pondweed)[61]. The $C_{19}$ to $C_{24}$ n-alkanes, maximising at $C_{20/21}$, in samples from Scottish and Lithuanian sites are consistent with the submerged aquatic plant genus *Nymphaea*[61] (Table 1). Additionally, an n-alkane proxy $P_{aq}$, gives values of 1.0 for all five samples where a $P_{aq}$ could be calculated, also indicating a submerged/floating aquatic plant[61]. Furthermore, the carbon preference indexes gave values between 1.10 and 1.20 (see Supplementary Information), which is consistent with a (biodegraded) submerged aquatic plant[56] (Table 1). The rhizomes in these plants have a far less pronounced odd-over-even carbon preference than the leaves[59], which is likely to reflect their sedimentary microenvironment, where they are far more exposed to biomolecular reworking and biodegradation from microbial inputs than leaves from the same plant. The n-alkanes in these calculus samples also display this unimodal 'hump' which is consistent with the consumption of rhizomes from a submerged aquatic plant.

## Brassicaceae

The $C_{27}$, $C_{29}$ and $C_{31}$ (trace) n-alkanes, with the $C_{29}$ n-alkane predominating, were identified in DL5838, Isbister. This is indicative of a higher plant wax origin from leafy greens[62]. Although these compounds are only present in minor amounts, the $C_{29}$ n-alkane is dominant in Brassicaceae epicuticular leaf waxes including *Brassica oleracea* (cabbage)[63] (Supplementary Information pp.54–55) and *Coincya* spp[64]. and has been used to identify *Brassica* leaves (cabbage) from archaeological contexts[56,63]. In the modern and archaeological studies the $C_{29}$ n-alkane accounted for over 90% of the hydrocarbon fraction, with the $C_{29}$ n-alkane/total n-alkanes ratios being: ~95:5 and ~96:4 for *Brassica* and *Coincya* respectively; this is consistent with the ratio of 96:4 for DL5838. Domesticated *Brassica* spp. vegetables such as cabbages and turnips are unlikely to have existed in Neolithic Orkney; however, sea kale (*Crambe maritima*) is native (see Supplementary Information). It has a distinct blue-grey-green waxy (glaucous) appearance in contrast to the other native Brassicaceae[30], reflecting the significant amount of wax on its leaves, with the dominance of the $C_{29}$ n-alkane making the wax monolayers/sheets more uniform and so providing a more protective and less permeable barrier[64] consistent with the very high $C_{29}$ n-alkane predominance that has been linked to glaucousness in

**Table 2 | Compound identifications of the $C_1$ to $C_6$ alkylpyrroles**

| Compound | LRI[a] | Mass spectral Characteristics | Dist[b] Cave 1 | Quant[b] 249-01 | Quant[b] 2552-02 | Isbist[b] DL715 | Isbist[b] DL5119 | Isbist[b] DL5838 | Mont[42,c] kerogen | Bili[42] | Chlor a[42] |
|---|---|---|---|---|---|---|---|---|---|---|---|
| 1. 2-methylpyrrole | 841 | 81(70), 80(100), 53(60) | √ | √ | √ | √ | √ | √ | | √ | √ |
| 2. 3-methylpyrrole | 850 | 81(70), 80(100), 53(50) | √ | √ | √ | √ | √ | √ | √ | √ | √ |
| 3. 2,5-dimethylpyrrole | - | | - | - | - | - | - | - | | (√) | - |
| 4. 2,4-dimethylpyrrole | 930 | 95(62), 94(100), 80(50) | √ | √ | √ | √ | √ | √ | √ | √ | √ |
| 5. 2,3-dimethylpyrrole | 939 | 95(75), 94(100), 80(40) | √ | √ | √ | √ | √ | √ | √ | √ | √ |
| 6. 3,4-dimethylpyrrole | 953 | 95(60), 94(100), 80(30) | - | √ | √ | √ | √ | √ | √ | √ | √ |
| 7. 2-ethyl-4(?)-methylpyrrole | 1007 | 109(30), 108(3), 94(100) | √ | √ | √ | √ | √ | √ | | (√) | (√) |
| 8. 2-ethyl-3(?)-methylpyrrole | 1010 | 109(40), 108(5), 94(100) | √ | √ | √ | √ | √ | √ | | √ | √ |
| 9. 4-ethyl-2-methylpyrrole | 1016 | 109(37), 108(5), 94(100) | √ | √ | √ | √ | √ | √ | | (√) | (√) |
| 10. 2,3,5-trimethylpyrrole | 1020 | 109(62), 108(100), 94(37) | √ | √ | √ | √ | √ | √ | √ | √ | √ |
| 11. 3-ethyl-4-methylpyrrole | 1024 | 109(48), 108(7), 94(100) | - | - | - | √ | √ | √ | √ | √ | √ |
| 12. 2,3,4-trimethylpyrrole | 1041 | 109(68), 108(100), 94(33) | - | - | - | √ | √ | √ | √ | √ | √ |
| 13. ethyldimethylpyrrole | 1091 | 123(40), 108(100), 94(20) | - | - | √ | - | √ | √ | | (√) | (√) |
| 14. ethyldimethylpyrrole | 1099 | 123(35), 108(100), 94(5) | - | - | tr. | tr. | - | tr. | | (√) | (√) |
| 15. 4-ethyl-2,3-dimethylpyrrole | 1103 | 123(37), 108(100), 93(6) | √ | √ | √ | √ | √ | √ | √ | √ | √ |
| 16. 3-ethyl-2,4-dimethylpyrrole | 1109 | 123(40),122(30),108(100) | √ | √ | √ | √ | - | √ | √ | √ | √ |
| 17. 2,3-diethyl-4-methylpyrrole | 1181 | 137(30),122(100),107(50) | - | - | - | - | - | tr. | | √ | (√) |
| 18. 3-ethyl-2,4,5-trimethyl-pyrrole | 1199 | 137(35),122(100),107(15) | tr. | - | - | - | tr. | tr. | √ | √ | √ |
| 19. 3-methyl-4-isobutyl-pyrrole[d] | 1130 | 137(10),122(10),94(100) | tr. | tr. | tr. | - | - | tr. | n/a | - | - |
| 20. 2,3-dimethyl-4-n-pro-pyl-pyrrole[d] | 1185 | 137(25),122(35),108(100) | - | - | - | tr. | - | tr. | n/a | - | - |
| 21. 3-ethyl-4-isobutyl-pyrrole[d] | 1239 | 151(5),136(35),121(70), 109(55),94(100) | tr. | - | tr. | - | - | tr. | n/a | - | - |
| 22. 3-methyl-4-neopentyl-pyrrole[d] | 1258 | 151(5),94(100) | - | - | - | tr. | - | tr. | n/a | - | - |
| 23. 2-methyl-3-ethyl-4-n-propylpyrrole[d] | 1258 | 151(5),122(100),108(90) | - | - | - | - | - | tr. | n/a | - | - |

√ = present; - = absent/not detected; (√) = present, but only as a minor component (<1%); tr. = compounds identified as trace constituents (based on retention times and mass spectra) from those samples with $C_1$ to $C_{5+}$ alkylpyrroles thermolytically-derived by pyrolysis-gas chromatography mass spectrometry at 610 °C (following thermal desorption/extraction of free compounds at 310 °C) and comparisons with alkylpyrroles from pyrolysed Monterey kerogen and tetrapyrrole model compounds[42].

Mont kerogen Monterey kerogen, *Bili* bilirubin (an animal-derived tetrapyrrole pigment), *chlor a* chlorophyll a (a plant, algae and cyanobacteria-derived tetrapyrrole pigment), *n/a* not applicable.
[a]LRI = linear retention index.
[b]Calculus samples: Distillery Cave 1; Quanterness samples 249-01 and 2552-02; Isbister samples DL715, DL5119 and DL5838.
[c]Only major alkylpyrroles reported in the original publication[42].
[d]Alkylpyrroles in bold derive from the C8 side chain in bacteriochlorophylls c and d found in green bacteria such as Chloroflexus sp[45]. and known to naturally occur in association with red seaweeds such as Porphyra umbilicali[4,46,47].

Brassicaceae leaf wax[64]. This protects against light, temperature and wind, with an increase in wax production linked to significant salt exposure[65], consistent with *Crambe maritima*'s habitat of shingle beaches. In Orkney, *Crambe maritima* is found only on the beaches of South Ronaldsay, adjacent to Isbister (see Supplementary Information p.55).

**Beeswax/propolis wax**
Chemical evidence consistent with a biodegraded/archaeological beeswax[41] or propolis wax[66] was also detected at Cabeço da Arruda (Table 1 and Supplementary Information) adding to the evidence for its widespread use across Europe in prehistory. Honeybee exploitation is illustrated on some Spanish Mesolithic rock art[67] and it was identified as a component of hafting material in the Late Upper Palaeolithic period[68] and as a dental filling in a Neolithic human tooth, also from southern Europe[69]. Biomolecular evidence has also been found widely on Neolithic sites across Europe[41].

## Discussion

The widespread perception of a rapid transformation from marine-based to terrestrial diets at the transition between the Mesolithic and the Neolithic[36] is due primarily to the dominance of carbon and nitrogen (C&N) stable isotope analysis in dietary reconstruction. Yet, in all the studied populations that have been analysed using C&N stable isotope analysis, a re-examination of the raw data is entirely consistent with the biomolecular evidence for consumption of seaweed or freshwater macrophytes (Supplementary Information). Aquatic ecosystems are complex with some isotopic variability even within the same species of seaweed[11,70,71], freshwater algae[72], and freshwater aquatic plants (macrophytes)[73], making use of this C&N isotopic data to detect these resources somewhat challenging[72,74], certainly where it represents <20% of the diet. Moreover, while stable isotopic analysis remains valuable in providing a broad view of major foods consumed, this is dependent on the dietary protein intake, with seaweeds lower in protein

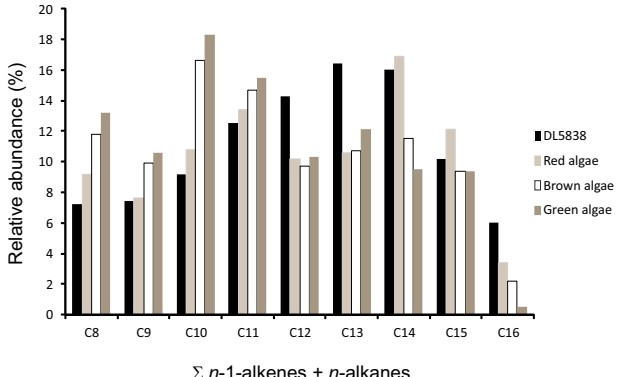

**Fig. 4 | Comparison of the sum of the *n*−1-alkene + *n*-alkane distributions (Σ *n*−1-alkene + *n*-alkane) (C$_8$ to C$_{16}$ carbon numbers) of calculus sample DL5838 from Isbister (highest organic content and most abundant macroalgal biomarkers) and modelled relative abundances predicted for red, brown and green macroalgae (seaweed).** These are based on their fatty acid composition and subsequent oxidative polymerisation and defunctionalisation, as observed in archaeological and ancient biomacromolecules−'chemical fossils'−in previous studies[40,51,52]. Source data are provided as a Source Data file.

compared to meat or fish, and can therefore be overlooked due to equifinality[75], where other foods can, in combination, be used to explain the stable isotopic data observed, sometimes in contradiction to other archaeological evidence available[75]. Our results therefore raise questions over the uncritical use of C&N stable isotope data in some studies, currently the dominant force in paleodietary reconstruction, and suggest this research may offer a powerful additional and complementary scientific approach. Recovery and identification of characteristic biomolecules in dental calculus is one of the few ways to obtain direct evidence of ingestion of plant species in prehistoric populations and can be used to identify specific items that may not be visible otherwise.

La Corona is a Mesolithic site (6059–5849 cal BC) in southeast Spain[76] around 80 km from the coast, today. Seaweed consumption so far inland may be surprising, but the Mesolithic was a population focused on aquatic resources and likely to have moved around using waterways while higher sea levels during the mid-Holocene are likely to have made the distance to the coast shorter[77]. Marine shellfish, found at the site[76], supports the connection with the sea. Although the consumption of marine resources was discounted in a C&N stable isotopes analysis[78], the wide variability for both δ$^{13}$C and δ$^{15}$N that is evident in the samples is consistent with seaweed consumption[11,70,71] (Supplementary Information pp.151–154). The studied individual from the Mesolithic Lithuanian site of SA5 and all three individuals from the Mesolithic Portuguese site of Cabeço do Pez produced evidence of consumption of submerged freshwater aquatic plants (macrophytes). Cabeço do Pez is one of a suite of late Mesolithic shell middens that contain large number of human skeletons in the Muge and Sado valleys. Dietary analysis based on C&N isotopes from this population suggested a mixed terrestrial diet and some marine protein;[79] however, it is likely the stable isotope data for submerged aquatic plants was not originally identified.

Biomolecular evidence for consumption of seaweed was found in almost every Neolithic sample from Orkney, and from Distillery Cave, Oban, Scotland. Isbister and Quanterness, Orkney, are chambered cairns (tombs) located in one of Europe's richest Neolithic landscapes. They contained extensive evidence for wild resources including inshore and deep-water fish, birds and wild plants in addition to agricultural produce. Despite this, the C&N stable isotope values of human bones suggest a primarily terrestrial diet[35,80] at both sites. In fact, the variability in the C&N isotopes, in particular of red seaweeds, can have

notably lower δ$^{13}$C values outside the range normally considered 'marine'[11,70,71] (Supplementary Information p.68), and may have been misidentified or undetected. The C&N values for the red seaweeds *Palmaria palmata* (dulse) (δ$^{13}$C = −16.5 to −22.0‰) and *Porphyra umbilicalis* (δ$^{13}$C = −19.5 to −21.7‰) in Scottish waters[71] are within the range of the human collagen values, after adjustment for trophic levels for each isotope ratio (δ$^{13}$C ~ 1‰ and δ$^{15}$N ~ 3.5–5‰)[53], and are therefore consistent with significant red seaweed consumption, with some input from other marine sources[79].

Three Chalcolithic/early Bronze Age samples from inland sites in Scotland and two Early Middle Age (5–6 C AD) samples from Lithuania have biomarker evidence for submerged freshwater aquatic plant consumption, suggesting their exploitation as food in Europe extended over at least two millennia into the Middle Ages, despite their use rarely being considered or recognised.

Seaweed has been suggested as human food in antiquity before[81] and consumption of marine resources is expected in the Mesolithic. However, the consistency and abundance of our evidence, (22 of 37 calculus samples), suggest that the use of seaweed as human food was widespread in Europe during the Mesolithic, into the Neolithic with evidence for freshwater aquatic plants extending well beyond this, highlighting some of the complexities embedded in the broad move away from use of wild resources to agricultural dominance that began in the Neolithic period. In most cases, the freshwater macrophytes are present in inland sites, while seaweed is present in coastal sites; the only exception being La Corona though, as a Mesolithic site, this fits well with the known marine focus of this time period. More broadly, the exploitation of freshwater aquatic plants that is evident throughout our study period across a wide geographical area, can be reasonably explained by the customary use of wild resources to supplement agricultural produce in subsistence populations, something that continues to be common today, either as preference or by necessity, since the one does not eliminate or compromise the other.

Today, seaweed and freshwater aquatic plants are virtually absent from traditional, western diets and their marginalisation as they gradually changed from food to famine resources and animal fodder, probably occurred over a long period of time, as has also been detected elsewhere with other plants[82]. Our study therefore also highlights the potential for rediscovery of alternative, local, sustainable food resources that may contribute to addressing the negative health and environmental effects of over-dependence on a small number of mass-produced agricultural products that is a dominant feature of much of today's western diet, and indeed the global long-distance food supply more generally.

Here, we have provided evidence that seaweed and freshwater aquatic plants were chewed and therefore most probably ingested in the Mesolithic and Neolithic periods, while freshwater aquatic plants were also ingested in the Bronze Age, and Early Middle Ages. Over 70% of the samples where biomolecular evidence survived had evidence for ingestion of red, green or brown seaweeds, or freshwater aquatic plants, with one sample from Orkney also containing evidence for a *Brassica*, most likely sea kale. This evidence occurs in different places across Europe, from southern Spain to Orkney in north Scotland. There has been little archaeological evidence for seaweed and freshwater aquatic plants to date, most likely due to their degradation over archaeological times periods. However, while these have gone unrecognised in C&N stable isotope studies, possibly due to the complexities in their identification, all the C&N stable isotope studies of these samples that have been carried out are consistent with our results. The switch from a predominance of wild to domesticated resources was likely gradual while the exploitation of wild resources, from mushrooms to shellfish and seaweed and wild haymaking, still endures in places across Europe, today.

## Methods

Permission was granted from the National Museums of Scotland for all the material from Scotland except Isbister, that was provided by The Orkney Museum; from the osteological collection, Faculty of Medicine, Vilnius University, for all Lithuanian samples; Schipluiden and Hardinxveld samples were provided by Provinciaal Archeologisch Depot Zuid-Holland, Alphen aan den Rijn, Netherlands. For Iberian sites these were provided by the following projects: CGL2008-03368-E, CGL2009-07572-E/BOS, Spanish Ministry of Education and Science. All other samples were provided by the excavation site directors.

Sequential TD-GC-MS and Py-GC-MS facilitates the identification of both free/unbound and polymerised/bound organic components[40,83,84], which can be present in many archaeological organic residues including the organic component in dental calculus. Thermal desorption coupled with gas chromatography-mass spectrometry is a rapid and direct method for the identification of free biomarkers in a broad range of organic materials[40,83,84]. Thermal desorption is effectively instantaneous, making it time efficient, and requires minimal sample preparation which reduces the likelihood of contamination and sample loss in comparison with conventional GC-MS, yet thermally extracts a wide range of organic compounds, rather than achieving this using organic solvents such as chloroform/methanol (2:1 v/v) or dichloromethane/methanol (2:1 v/v). It also requires very small sample sizes (<0.1 mg of organic residue), allowing the virtually non-destructive analysis of often precious archaeological and/or museum samples[40,84]. Crucially, it can be conveniently combined with Py-GC-MS, which involves the release of 'bound' organic material and the decomposition of macromolecular/polymeric organic material using heat to yield low molecular weight products characteristic of the original macromolecule/biopolymer and sufficiently volatile to then be separated and identified by GC-MS[40,85]. Since the calculus samples contain a 'chemical fossil' component, which is a biopolymer deriving from the lipids, proteins and carbohydrates in the original organic natural products (seaweed/freshwater algae and freshwater plants) the Py-GC-MS analyses, in particular, are not directly comparable with the free biomarkers/biomolecules present in the extant organisms (e.g., seaweeds) and therefore modern reference materials for comparison are not appropriate. However, using free and bound/polymeric biomarkers characteristic of the original organic materials this analytical approach has been applied successfully to a number of studies on archaeological dental calculus[62,82,86–88]. Sequential TD-GC-MS and Py-GC-MS. TD/Py-GC-MS analysis was performed on a CDS Pyroprobe 2000 (Chemical Data System, Oxford, PA, USA) via a CDS1500 valved interface (320 °C) linked to a Hewlett-Packard 5890 Series II gas chromatograph fitted with a split/splitless injector (280 °C), interface to a Trio 1000 mass spectrometer (electron voltage 70 eV, filament current 220 uA, source temperature 230 °C, interface temperature 325 °C). The MS was set to scan in the range 40–850 amu. The dental calculus sample (0.5–20 mg) was weighed into a quartz tube with glass wool end plugs. The tube was placed into a pyroprobe platinum heating coil and sealed into the valved interface. The TD/Py temperature was held for 10 s. The samples were thermally desorbed at 310 °C, followed by pyrolysis at 610 °C. Separation was performed on a fused silica capillary column (30 m × 0.25 mm i.d.) coated with 0.25 μm 5% phenyl methyl polysiloxane (DB-5) stationary phase. Initially the GC oven was held at 35 °C for 5 min and then temperature programmed from 35 °C to 320 °C at 6 °C min and held at final temperature for 15 min, total 67.5 min, with Helium as the carrier gas (flow 1 mL/min, initial pressure of 45 kPa, splitless injection 1 min.). Peaks were identified on the basis of both their mass spectra NIST Mass Spectral Database and additional referenced data (see also Supplementary Information), and relative retention times (relative retention indices). Blanks were run between all samples (post-Py-GC-MS) to ensure there was no carryover between samples and all organic compounds observed derived from the calculus samples analysed. Blanks were also run prior to analyses and post-analyses to determine any potential laboratory or instrumental contamination.

### Reporting summary

Further information on research design is available in the Nature Portfolio Reporting Summary linked to this article.

## Data availability

All information on the samples and the data generated and analysed in this study are included in the manuscript and the Supplementary Information and Supplementary Data file. This includes the archaeological and biomolecular data necessary for this research. Source data are provided as Source Data files. Samples were provided from the following locations: Orkney Museum, National Museums Scotland, Faculty of Medicine, Vilnius University, Faculty of Biosciences, Universitat Autònoma de Barcelona, Provinciaal Archeologisch Depot Zuid-Holland (Provincial Archaeological Depot South Holland) in Alphen aan den Rijn; Institute of Archaeology and Ethnology Polish Academy of Sciences, Cultural Heritage Foundation, Västerås, Sweden. No data has been reused in this work. Source data are provided with this paper.

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

## Acknowledgements

Funding was provided by the I + D MICINN, (project code HAR2012-3537) and the Orkney Archaeology Society. The Kanaljorden excavation-project at Stiftelsen Kulturmiljövård, Sweden, provided funding for the analysis of the Kanaljorden-samples. Sheila Garson facilitated sample collection in Orkney. Orkney Museum and Anders Fischer are thanked for providing samples. Lorraine McEwan, University of Glasgow, produced Fig. 2 using Corel Draw v23. Analytix and CDS are thanked for their technical advice. Professor Joann Fletcher is also thanked for technical assistance with the manuscript.

## Author contributions

KH conceived and led the project and wrote the paper, SB conducted the biochemical analyses and wrote the paper. FH, LKB, ZM, IS-T, AS, MES provided samples, contributed to the main text and SI.

## Competing interests

The authors declare no competing interests.
