## [Peer Review File · Nature Communications]

Human consumption of seaweed and freshwater aquatic plants in ancient EuropeReviewers' Comments:

Reviewer #1:

Remarks to the Author:

This is a timely article concerning human consumption of seaweed and freshwater aquatic plants in Europe. The results in this article are that of 37 calculus samples with identifiable chemical biomarkers, 26 samples showed evidence of seaweed, freshwater algae and aquatic plants. The methodology is appropriate and novel, the interpretations appear mostly sound, and the conclusions justified. This work has broad implications for our understanding of past European diets.

General comments:

The composition of the Monterey kerogen is described in the publication 42 as follows "the relatively high amounts of tetrapyrrole pigments may be related to the abundant algal contribution received by both sediments or to unique preservation characteristics associated with these deposits."

However, in the current manuscript, this suggestion is taken as fact (e.g. in Table 42, the kerogen is called "fossil algae"). The composition of the Monterey kerogen should be discussed further before it is used as a comparative reference. Even in the supplementary information, this is not discussed.

Including the supplementary information, this is an incredibly long article, yet there is no section anywhere generally explaining the employed methods (how/why they work) that is accessible to most archaeologists. This should be rectified, or a good review covering TD- and Py-GC-MS (that does not expect the readers to be chemists) should be cited, and the manuscript (or supplementary information) generally be made more accessible to those not yet versed in pyrolysis-GC-MS. It would also be worth discussing e.g. why there is little sense in analysing seaweeds/freshwater plants directly to gain a reference to compare to, and what time period calculus reflects (e.g. last year of life, several years) and to what extent seaweed/freshwater plant consumption is quantifiable by GC-MS.

If journal guidelines permit, I would appreciate a separate conclusion section summarising the main findings and implications.

Other suggested changes (line-by-line):

Line 11 the Mesolithic involving "6000 years of intensive coastal and marine resource exploitation" is not the case throughout all of Europe, this statement should be made more generally valid

Line 31-32 The reference for the poem attributed to St Columba on collecting dulse off rocks is not sufficient. Depending on the source this poem sometimes also simply mentions "seaweed" instead of rocks, and it is unclear if this poem really originates from St Columba's time, or was simply attributed to him. If you can find a better reference that would be great (I was unable to a few years ago), or you could fix it in the manuscript by writing that this poem was attributed to St Columba, rather than that it was by St Columba.

Figure 1 Please improve the resolution

Lines 118 and 186 I have been unable to find a reference confirming that Chloroflexus is not found in association with brown intertidal seaweeds. The reference 44 cited here does not say that Chloroflexus sp. is known to occur specifically with intertidal red seaweeds, but rather with seaweeds in general. In my opinion, these references aren't sufficient to show that the presence of bacteriochlorophylls c and d are indicative of red seaweed consumption in particular. (also note that in line 116 bacteriochlorophyll is misspelled, as well as in the supplementary information lines 1191, 1610, and 1791)

Table 2 mentions bilirubin, but why it is relevant is not explained in the manuscript or the SI.

Line 157 please specify that this is dry weight (I assume)

Figure 3 Please make this figure more easily readable when printed in grey-scale, and add that DL5838 is a sample from Isbister to the description

Line 198 this heading "algae" is confusing, considering the previous heading "seaweed", since seaweeds are macroalgae

Line 196 please give more information on how the relative abundances for red brown and green macroalgae were modelled

Line 202 algae are discussed here, but the reference 56 is for seagrass. Seagrasses are not algae, they are higher plants.

Line 258 and following: "C/N stable isotope analysis" is inaccurate, since C/N refers to the ratio of C to N. I suggest writing "C&N stable isotope ratio analysis" instead.

Line 263 recent results from modern animals have shown that significant seaweed consumption can easily be identified in herbivores (sheep) using carbon isotope ratio analyses of bone collagen, bone apatite or tooth enamel. It is only in omnivorous (particularly human) diets, where seaweed constitutes a smaller extent of the diet, that identification of seaweed consumption is difficult. Isotopic variability between seaweeds of the same species are not such a large issue as painted in the manuscript: where isotopic variation is not systematic (so not e.g. only juveniles, or seaweeds at specific locations have lower isotope ratios), the mean is what affects the consumer's isotope ratio, since people in the past could not selectively pick only those individuals of a species that have comparatively high or low isotope ratios. Interindividual variation is thus not an issue if means are consistent. Additionally, significant isotopic variability between seaweed species is not that problematic, considering not all seaweeds are commonly consumed (and would not have been consumed in the past, e.g. chalky seaweeds are not particularly tasty or comfortable to eat, and some of the epiphytes are much too small and rare to have constituted a relevant part of past diets). Commonly consumed seaweeds like *Laminaria* spp, *Ulva* spp., *Palmaria palmata*, *Fucus* spp., *Himantalia elongata* etc. have generally elevated $\delta^{13}\text{C}$ values compared to terrestrial C3 plants (two of the few exceptions are *Porphyra* spp., and *Chondrus crispus* which are commonly eaten and have $\delta^{13}\text{C}$ values similar to terrestrial C3 plants). I still agree identifying seaweed consumption is difficult using stable isotope ratios though, because when diets contain less than ca. 20 % "marine" protein, this tends not to be identifiable due to isotopic variation in terrestrial plants, and also because it isn't possible to distinguish fish + terrestrial plants vs seaweed (other than *Porphyra* spp. and *Chondrus* c.) + meat using stable isotope ratios due to equifinality.

Line 274 "correlates with seaweed consumption" - Perhaps a better wording would be "consistent with seaweed consumption", since this is not a correlation in the statistical sense

Line 288 the term "depleted" is true for both high and low $\delta^{13}\text{C}$ values if it is not specified which isotope is depleted (^{12}C or ^{13}C). I'd suggest "can have notably lower $\delta^{13}\text{C}$ values"

Lines 290 and 291 "%o" signs missing

Line 292 please add the word "ratio" before the brackets

Line 297 the manuscript says "suggesting this practice [of freshwater plant consumption] certainly extended into the Middle Ages". This implies continuity, but since this concerns three samples from Scotland and two from Lithuania, there is no continuity, since these places are so far apart, that they had little relation to each other. Whether or not there was an ebb and flow in freshwater plant consumption, or whether this was mostly continuous until it fell out of favour recently requires much further research.

Line 299 I think it is worth quantifying your results here – 26 of 37 calculus samples suggest marine/freshwater plants/algae as human food. Also, considering the wording: only green seaweeds are technically plants

Line 307 RE use of wild resource to supplement agricultural produce - this was common everywhere in the Neolithic, considering domestic animals consumed mostly wild graze and browse

Line 316 Other strong arguments for seaweed consumption are that a) seaweed does not require irrigation, b) it does not require space on land to grow c) it does not require fertiliser. So sustainable seaweed farms end up being better for the environment than e.g. a wheat field.

Line 319 this is not a grammatically complete sentence

Supplementary Information

The supplementary information has 199 pages, with 5333 lines. A simple “see also SI” or referencing “SI” in the manuscript is not that helpful, since finding the relevant section is difficult considering the sheer length of the SI. This has made reviewing this manuscript very arduous as well (in addition to its length). Please give page or paragraph numbers when referring to the supplementary information (if giving paragraph numbers, please include more numbered headings in the supplementary information). The supplementary information also often repeats itself and the manuscript word by word, please shorten it or put more headings in the table of contents and include hyperlinks. I would also prefer one long reference list at the end including all references rather than having several since this involves a lot of page turning to find the correct reference list.

Line 1297 and 1302 add “‰”

Line 4025 “it should be noted that if any marine component was fish or shellfish it would be expected that the isotopic data would reveal this.” Only if this consumption was larger than ca. 20 % of dietary protein when averaged over all seasons. Smaller amounts of marine food consumption are commonly not identifiable.

Line 4029 If large amounts of e.g. kelp were being consumed (let’s say 50 % of the diet) throughout life, this would most certainly be visible in $\delta^{13}\text{C}$ values. See publications on modern seaweed-eating Orkney sheep. Seaweed consumption might be confused for fish consumption though. I fully agree with your disagreement in line 4032 with the paper’s statement “...this population did not have any direct and regular dietary contact with the sea”, since $\delta^{13}\text{C}$ values cannot be used to exclude marine food consumption since minor amounts would not be visible, and because some marine foods have $\delta^{13}\text{C}$ values similar to terrestrial C3 plants.

Line 4038 considering only one individual had equivocal evidence of marine food consumption, it might be a bit much to conclude the entire population at La Corona had dietary contact with the sea. A larger sample size would be needed for such a general assertion

Line 4382 I do not have access to this reference, but generally elemental data on bone is not reliable due to diagenesis and should be discounted (enamel would be ok for Sr, Ba and Ca). See also BURTON, J.H. & T.D. PRICE. 2006. The Use and Abuse of Trace Elements for Paleodietary Research, in Biogeochemical Approaches to Paleodietary Analysis: 159–71. https://doi.org/10.1007/0-306-47194-9_8

Line 4386 I think there is a typo here – what is ORSr?

Line 4588 "extent"

Reviewer #2:

Remarks to the Author:

The paper shows many analyses reporting chemical markers extracted from human dental calculus often related to compounds present in vegetable plants or animals. The results demonstrate that in Europe, in the Mesolithic - Early Middle Ages period, there was a widespread consumption of coastal resources (mainly seaweed and submerged aquatic plants). The paper may be accepted after minor revisions.

A general comment on biomarker. The authors report "...identifiable and characteristic biological marker ('biomarker') compounds extracted from..." It is not a biological marker but a chemical compound contained in a sample and, in this case, related to a food. The term biomarker as reported in the literature seems to be a fashion of the past! Anyway, biomarker can be accepted but please correctly define it.

Figure 2 at page 9. In the caption: Figure 13? Figure 2 is too much complex, I suggest to simplify reporting almost three reconstructed pyrograms with peak recognition

Page 12

Lines 153-156. Pyrograms report the presence of decomposition/rearrangement products of aminoacids: this discussion is misleading. DKPs are present both for animal and vegetable proteins and it is hardly understandable to assess their origin unless to perform aminoacid analysis or better proteomic. Moreover, depending on pyrolysis temperature you may produce different molecules. Alkylpyrroles are often related to diary products: how can you rule out their presence?

Lines 160-163. Because fatty acids are not volatile, you cannot observe them in TD-GC-MS as well as in PY-GC-MS. Most probably they are not eluted, therefore you cannot sustain "...confirming an absence of these free lipid biomolecules". The sequence alkane/alkene of algal biopolymers should be demonstrated in your experimental conditions and compared with literature data: it is not clear if algae data reported in Figure 3 are obtained by the authors.

Supplementary data

The paper is well constructed, but too many results are discussed in the supplementary data with several repetitions to explain the presence of some "biomarkers". I suggest summarizing all the identified molecules per each sample and site in one table and avoid comments in the supplementary file. The figures are hardly understandable due to the many symbols. Please, simplify CO₂ and acetonitrile could be the result of a laboratory pollution: have you daily checked the blanks?

Reviewer #3:

Remarks to the Author:

This is an exciting paper, which offers evidence to critically evaluate the orthodoxy that there was a shift away from the use of marine resources in northwest Europe during the Neolithic. This orthodoxy is based on C/N isotopic analyses of skeletal populations, that can only offer broad brushstroke analyses of past diets and often underestimate the use of plants and algae. This paper instead uses TD-GC-MS and Py-GC-MS to investigate dental calculus from Mesolithic, Neolithic and Bronze Age and early Medieval skeletal remains, and demonstrates the ongoing consumption of seaweed into the Neolithic, as well as the use of freshwater plants into early Medieval times. The work is of significance to our understandings of the spread of agriculture in Europe. It is also significant methodologically, describing a pathway to better analyse the consumption of plants and algae in past diets. I am not a specialist in biomolecular archaeology and cannot comment on the merits of the chemistry used in this study. I can, however, comment on the research design and interpretation.

I have two minor comments:

1) At times the authors present an overview of an "agricultural revolution" that does not reflect the

nuance now understood in global agricultural origins research. For example, "This represented a dramatic shift from all previous human existence which was based on hunting, gathering and non-accumulation, and established the social and economic foundations that underpin today's world, including the control and management of terrestrial food sources, land ownership, storable surplus, population increase and full sedentism," (lines 50-56). Whilst this was a significant point in human history, this sentence ignores the many "hunter-gatherer" economies that did store surpluses, and managed their landscapes and food resources, and the agropastoral economies that were not sedentary, etc. A little nuance here will go a long way.

2) There seems to be two stories presented in the paper, one about seaweed and another about freshwater aquatic plants. I believe these are not mutually exclusive stories and should be in the same paper. However, with the way parts of this is written the two claims seem to not be clearly demarcated, leading to some miscommunication. Line 299-301, "Three Chalcolithic/early Bronze age samples from inland sites in Scotland and two early Middle Age (5-6C AD) samples from Lithuania have biomarker evidence for submerged freshwater aquatic plant consumption, suggesting this practice certainly extended into the Middle Ages." As this follows from a paragraph on the consumption of seaweed and is directly followed by, "Seaweed has been suggested as human food in antiquity before and consumption of marine resources is expected in the Mesolithic," it took me several reads to understand that "the practice extending into the Middle Ages" was not seaweed use. I suggest a small rewrite here and in the introductory paragraph where it feels like freshwater aquatic plant use is thrown in as an afterthought and it is confusing for the reader to follow and distinguish between the claims made.

Also, it should be Northwest Europe and Southwest Asia, not North West or Southwest.

Overall, this is an exciting paper that changes our picture of past diet and also has thoughtful forward focus.

RESPONSE TO REVIEWERS

REVIEWER COMMENTS

Reviewer #1 (Remarks to the Author):

This is a timely article concerning human consumption of seaweed and freshwater aquatic plants in Europe. The results in this article are that of 37 calculus samples with identifiable chemical biomarkers, 26 samples showed evidence of seaweed, freshwater algae and aquatic plants. The methodology is appropriate and novel, the interpretations appear mostly sound, and the conclusions justified. This work has broad implications for our understanding of past European diets.

General comments:

The composition of the Monterey kerogen is described in the publication 42 as follows “the relatively high amounts of tetrapyrrole pigments may be related to the abundant algal contribution received by both sediments or to unique preservation characteristics associated with these deposits.”

However, in the current manuscript, this suggestion is taken as fact (e.g. in Table 42, the kerogen is called “fossil algae”). The composition of the Monterey kerogen should be discussed further before it is used as a comparative reference. Even in the supplementary information, this is not discussed. –

Reply: We agree this needed further discussion and following this very helpful suggestion have included it in the main text and repeated this in the SI for clarity. We have also amended the “fossil algae” to “Monterey kerogen” to reflect the more equivocal interpretation of the tetrapyrrole pigments-derived biomarkers in the Sinninghe Damste et al. 1992 research and have better contextualised the Monterey kerogen (including additional geological and geochemical data, providing additional evidence for the algal input and likely algal origin of the alkylpyrroles) and the alkylpyrrole biomarkers identified in the dental calculus, separately and in connection with each other, in the expanded section now, very reasonably, advised by the reviewer here.

Including the supplementary information, this is an incredibly long article, yet there is no section anywhere generally explaining the employed methods (how/why they work) that is accessible to most archaeologists. This should be rectified, or a good review covering TD- and Py-GC-MS (that does not expect the readers to be chemists) should be cited, and the manuscript (or supplementary information) generally be made more accessible to those not yet versed in pyrolysis-GC-MS.

Reply: We believe this is a helpful suggestion given the relatively ‘niche’ and specialised area of TD/Py-GC-MS and have amended and significantly expanded the main manuscript Methodology accordingly, with additional information and key references, providing a more detailed explanation of the methodology for non-experts such as archaeologists.

It would also be worth discussing e.g. why there is little sense in analysing seaweeds/freshwater plants directly to gain a reference to compare to,

Reply: We acknowledge that this would be helpful and have added this to the Methods in the manuscript; the calculus samples contain a ‘chemical fossil’, which is a biopolymer deriving from the lipids, proteins and carbohydrates in the original seaweed/freshwater algae and freshwater plants and so are not directly comparable with the free biomolecules present in the extant organisms. This biopolymer/macromolecular organic matter can, however, be usefully compared to organic-rich

ancient sediments, such as the Monterey kerogen, which are commonly investigated using the organic geochemical 'biomarker approach' also used in many archaeological studies by the lead author of this submission and other leading academics in the field, such as Professor Richard Evershed, with whom the first author has worked and published previously. Notably, this study uses the exact same analytical instrumentation and methodology utilised in the Buckley, Stott & Evershed 1999 study and other similar archaeological studies (cited in the submission) by the authors of this manuscript since then. Moreover, it is interesting to note that in previous studies where Py-GC-MS has been applied to the analysis of modern seaweeds using similar pyrolysis temperatures (500/600°C) a notable biomolecular similarity is seen between the Py-GC-MS profiles of extant seaweeds and the organic component of the calculus samples in this study (Wang et al. 2013). More specifically, the Py-GC-MS analysis of extant seaweed has revealed the C₁₃ n-1-alkene and n-alkane in equal (and significant) abundance, yet the C₁₄ n-1-alkene in significantly higher quantity than both of these lipid markers, dominating the pyrogram, while the C₁₄ n-alkane is not reported as a constituent of the pyrolysate, suggesting it is minor or absent (Wang et al. 2013). Additionally, the same study revealed significant amounts of the protein markers phenol, 3-methyl phenol, benzenepropanenitrile, indole, 3-methylindole, and hexadecanenitrile; these findings on the Py-GC-MS of modern seaweed are also observed in the calculus samples containing other macroalgal-associated biomarkers, i.e. the highly specific alkylpyrroles. Unfortunately, the Wang et al. 2013 study contains transcription errors and so while what we say here is clear from a scientific perspective (this is obvious from the relative retention times of the reported organic compounds, etc.), we do not feel it would be a suitable reference for this manuscript. These easily spotted errors and poor interpretation of the biomolecular evidence in similar studies on the use of Py-GC-MS for the study of modern seaweeds, usually in connection with biofuel potential, means – very frustratingly – these studies are too compromised to be cited, requiring a reinterpretation of the data presented to make meaningful sense of them, which is obviously beyond our remit (or at least should be!) when it comes to the manuscript presented here.

and what time period calculus reflects (e.g. last year of life, several years) and to what extent seaweed/freshwater plant consumption is quantifiable by GC-MS.

Reply: Given the inherent anthropological and diagenetic uncertainties, it is not possible to know how representative of an ancient individual's life, or which part of that life, the results from the calculus are, beyond it being reasonable to infer that only commonly and regularly consumed foods, with their attendant resilient biomarkers, would have been likely to have been incorporated to the extent that they can be detected in ancient dental calculus by chemical analysis (TD- and Py-GC-MS). Occasionally ingested foods and other materials would not be expected to be observed. In analytical terms the TD/Py-GC-MS can be said to be semi-quantitative, but in the scientific and archaeological context of this study it is largely qualitative. While not providing a full picture due to the uncertainties mentioned above and biases related to preferential survival of particular free and bound/polymeric biomarkers, sequential TD-GC-MS and Py-GC-MS analyses can, and do, provide interesting and important chemical evidence representing significant parts of the diet (e.g. Hardy et al 2012 and Buckley et al 2014) and other materials ingested/inhaled (e.g. smoke/combustion markers; see Buckley et al., Scientific Reports 2021).

If journal guidelines permit, I would appreciate a separate conclusion section summarising the main findings and implications.

Reply: Since a conclusion is not allowed by Nature Communications, we have summarised the main results in a paragraph at the end of the Discussion.

Other suggested changes (line-by-line):

Line 11 the Mesolithic involving “6000 years of intensive coastal and marine resource exploitation” is not the case throughout all of Europe, this statement should be made more generally valid

Reply: We agree and have amended this accordingly. The Mesolithic evidence is strongly marine that is not disputed but we have removed the specific reference to ‘coastal’ and replaced this with ‘aquatic’ to include inland water systems.

Line 31-32 The reference for the poem attributed to St Columba on collecting dulse off rocks is not sufficient. Depending on the source this poem sometimes also simply mentions “seaweed” instead of rocks, and it is unclear if this poem really originates from St Columba’s time, or was simply attributed to him. If you can find a better reference that would be great (I was unable to a few years ago), or you could fix it in the manuscript by writing that this poem was attributed to St Columba, rather than that it was by St Columba.

Reply: We agree with the reviewer and have added in ‘attributed to’ St Columba.

Figure 1 Please improve the resolution.

Reply: We have provided a new Figure 1.

Lines 118 and 186 I have been unable to find a reference confirming that Chloroflexus is not found in association with brown intertidal seaweeds. The reference 44 cited here does not say that Chloroflexus sp. is known to occur specifically with intertidal red seaweeds, but rather with seaweeds in general. In my opinion, these references aren’t sufficient to show that the presence of bacteriochlorophylls c and d are indicative of red seaweed consumption in particular. (also note that in line 116 bacteriochlorophyll is misspelled, as well as in the supplementary information lines 1191, 1610, and 1791).

Reply: There are TWO key points to note here:

1. Reference 44 (Selvarajan et al., 2019) was used since it seemed to offer a useful reference connecting seaweeds with Chloroflexi, with the superscripted ‘SI’ providing the connection to red seaweed specifically. [*Supplementary Information contains the reference Miranda et al., PLOS ONE 2013, connecting Chloroflexi with the intertidal red seaweed, Porphyra umbilicalis, plus Longford et al., Aquatic Microbial Ecology 2007, which identified Chloroflexi in a subtidal red seaweed (although NOT in green seaweed in the same study).] This use of reference 44 in the original submission was purely due to the reference limit, which prevented our inclusion of the Miranda et al. 2013 paper, in particular, in the main text, which we would have otherwise included.*

However, having further scrutinised the Selvarajan et al. 2019 paper, thanks to the helpful comments of the reviewer here, we now realise that this was not the most appropriate choice of article in any event. The ‘...and Chloroflexi¹⁹⁻²².’ in the text of Selvarajan et al. 2019 article does not cite ANY research papers that found Chloroflexi (as opposed to other bacteria, which WERE identified in the references 19 to 22 cited) in ANY seaweed, although reference 17 of the same paper DOES identify Chloroflexi in RED seaweed, which suggests this reference was meant to have been included in the ‘...and Chloroflexi¹⁹⁻²².’ and was perhaps accidentally omitted during revision(s) prior to publication. Consequently, following both the helpful comments of the reviewer and the compromised reference originally used to make a more general point on Chloroflexi associated with seaweed, we have now

replaced reference 44 with Miranda et al. 2013, which not only connects Chloroflexi with seaweed, but, more specifically, intertidal red seaweed, i.e. *Porphyra umbilicalis* (found specifically on most of the blades at moderate abundance in the Miranda et al. study, with relevance as the most likely parts to have been consumed as food in this study), and therefore is now consistent with the main (amended) text and the scientific research in this area to date. We have also added the Longford et al. 2007 paper, which identifies Chloroflexi with a subtidal red seaweed.

We have also carried out additional research on bacteria in seaweeds, including reviews of research in this area. Where relevant, the findings have been discussed in the main text and cited. The most comprehensive review by Hollants et al. 2013 found Chloroflexi only in association with red seaweed, being absent from the green and brown seaweeds included in this fairly extensive review of 161 studies. Another review often cited is Singh and Reddy, FEMS 2014; however, although this mentions Chloroflexi in its second paragraph on p.214, NONE of the studies cited there identified Chloroflexi in association with seaweeds and of all those reviewed it was only identified in a study on red seaweed, i.e. Longford et al 2007, although, perplexingly, it was not reported in connection with this study in the Singh and Reddy 2014 review. Of the over 200 previous studies of green, brown and red seaweeds it has only been red seaweeds where Chloroflexi has been found in association in any significant abundance (e.g. Miranda et al. 2013) and so while this may change with future studies in this area, even on the basis of the alky pyrrole data alone (although SEE BELOW), it is consistent with the identification of moderately abundant Chloroflexi in association with red seaweeds, specifically the intertidal red seaweed *Porphyra umbilicalis*, in a similar temperate/cool northern hemisphere environment to western/northern Scotland. The main text and SI still reflect these realities, BUT we have now made that connection clearer with the removal of the Selvarajan et al references and its replacement with Miranda et al 2013 and, citations allowing, Longford et al 2007.

2. To acknowledge the reasonable concerns of the reviewer, we also do not entirely rule out the possibility that an intertidal brown (or green) seaweed could explain the alky pyrrole distributions in the dental calculus samples (at least as a contribution) and indeed bacteria of the phylum Chloroflexi have been identified in one scientific study, i.e. Tourneroche et al., *Frontiers in Marine Science* 2020, albeit at very low levels unlikely to explain the relative abundances of the more unusual c and d bacteriochlorophyll-related alky pyrroles in the calculus samples. IMPORTANTLY, however, the red seaweed identification is based not only on the presence of the c and d bacteriochlorophyll-related alky pyrroles, the tetrapyrrole precursors of which are known to occur in Chloroflexi, which are known to occur in association with intertidal and subtidal red seaweeds (Miranda et al 2013 and Longford et al 2007), but the relative distributions of the lipid-related biomarkers. These can be related to the relative proportions of the original fatty acids constituting the acyl lipids of green, brown and red seaweeds, which differ markedly, as we explain in the main text. It is this COMBINATION of the highly distinctive alky pyrroles and their identification in association with subtidal and intertidal red seaweeds AND the n-1-alkene/n-alkane distributions (from a kerogen-type biopolymer deriving from labile acyl lipids, following oxidative cross-linking [via autoxidation] of the unsaturated and saturated fatty acids [followed by decarboxylation and reduction of the hydroxyl groups]; see Buckley et al 1999, plus seminal references in this area below, two of which were cited in our submission) observed in the dental calculus samples, which can be correlated with the relative abundances of the original fatty acids in green (C_{16} and C_{18} polyunsaturated fatty acids, with C_{20} acids in minor abundance and C_{22} acids very minor or absent), brown (less abundant C_{16} acids, with C_{18} acids being major and C_{20} acids more significant than green) and red (C_{20} polyunsaturated acids dominating, with significant amounts of C_{22} acids) seaweeds (references: 1,45,50-52 in the original submission). It can be seen that the expected n-1-alkene/n-alkane distributions are most similar to red seaweeds.

De Leeuw, J.W., Versteegh, G.J.M. & Van Bergen, P.F. Biomacromolecules of plants and algae and their fossil analogues. *Plant Ecology* **189**, 209-233 (2006)

Versteegh, G.J.M., Blokker, P., Wood, G.D., Collinson, M.E., Sinninghe Damsté, J.S. & de Leeuw, J.W. An example of oxidative polymerisation of unsaturated fatty acids as a preservation pathway for dinoflagellate organic matter. *Organic Geochemistry* **35**, 1129-1139 (2004)

Gupta, N.S., Michels, R., Briggs, D.E.G., Collinson, M.E., Evershed, R.P. & Pancost, R.D. Experimental evidence for the formation of geomacromolecules from plant leaf lipids. *Organic Geochemistry* **38**, 28-36 (2007)

Gupta, N.S., Briggs, D.E.G., Collinson, M.E., Evershed, R.P., Michels, R., Jack, K.S. & Pancost, R.D. Evidence for the in situ polymerisation of labile aliphatic organic compounds during the preservation of fossil leaves: Implications for organic matter preservation. *Organic Geochemistry* **38**, 499-522 (2007)

Gupta, N.S., Cody, G.D., Tetlie, O.E., Briggs, D.E.G. & Summons, R.E. Rapid incorporation of lipids into macromolecules during experimental decay of invertebrates: Initiation of geopolymer formation. *Organic Geochemistry* **40**, 589-594 (2009)

De Leeuw, J.W. On the origin of sedimentary aliphatic macromolecules: A comment on recent publications by Gupta et al. *Organic Geochemistry* **38**, 1585-1587 (2007)

The error in the spelling of bacteriochlorophylls in line 116 in the main text and lines 1191, 1610 and 1791 in the supplementary information have been corrected and we thank the reviewer for pointing these out.

Table 2 mentions bilirubin, but why it is relevant is not explained in the manuscript or the SI.

Reply: Bilirubin is present in Table 2 to acknowledge its use as a model tetrapyrrole compound in the key Sinninghe Damsté et al publication on the Monterey kerogen, mentioned by the reviewer above. Interestingly, while we have not focussed on relative quantitative data in our manuscript, the animal-derived bilirubin alkylpyrrole profile resulting from pyrolysis-GC-MS (see Fig.5 in the Sinninghe Damsté et al paper) is a poorer fit with the Py-GC-MS alkylpyrrole profiles in the calculus samples compared to the biologically more relevant chlorophyll a alkylpyrrole profile, but we felt it sensible to include bilirubin in Table 2 to reflect important research in this area, rather than appear to be selective in our choice of model tetrapyrrole compounds for this submission. However, there could be an argument for its removal from Table 2 given the animal origin of bilirubin and we would not object to that, if so.

Line 157 please specify that this is dry weight (I assume)

Reply: Yes, this is dry weight and we have amended accordingly.

Figure 3 Please make this figure more easily readable when printed in grey-scale, and add that DL5838 is a sample from Isbister to the description.

Reply: We have added 'from Isbister' in the description of the calculus sample DL5838. We have also added '...subsequent oxidative polymerisation and defunctionalisation, as observed in archaeological and ancient biomacromolecules – 'chemical fossils' – in previous studies (Buckley et al 1999; Versteegh et al 2004; Gupta et al 2007).' to the figure caption, so that it better relates to, and connects with, the main text above relating to this. We believe Figure 3 in grey-scale is now easily

readable.

Line 198 this heading “algae” is confusing, considering the previous heading “seaweed”, since seaweeds are macroalgae.

Reply: We agree with the reviewer here and have amended it to ‘Freshwater algae’, reflecting the likely origin of the algae contained in these samples, given the location of the archaeological sites, i.e. inland sites in relatively close proximity to bodies of freshwater (lakes and/or rivers).

Line 196 please give more information on how the relative abundances for red brown and green macroalgae were modelled

Reply: We have now better explained the modelling in the main text (and Figure 3 caption) and provide more information on this in the SI.

Line 202 algae are discussed here, but the reference 56 is for seagrass. Seagrasses are not algae, they are higher plants.

*Reply: Indeed, seagrasses are higher plants, although both here (Viso et al Phytochemistry 1993) and in current studies (e.g. Chen et al, Frontiers in Microbiology, Vol.13, 2022) [macro]algae and seagrasses are discussed together as ‘Marine macrophytes (seagrasses and macroalgae)’ (Chen et al, Frontiers in Microbiology, Vol.13, 2022) and there remains no **absolute** consensus on precise terminology, meaning while although there is a **general** consensus that algae are neither plants nor animals (nor fungi, nor bacteria) they can be linked to plants in some scientific studies – and also see below on the reviewer’s own comment on seaweed types and any connection with plants. HOWEVER, this referenced paper (Viso et al 1993, Ref.56) includes what we regard as a useful in-depth chemical ‘compare and contrast’ of seagrasses with algae and on p.385 of the article it states: ‘...been observed in other seagrasses. This distribution differs from algae where alkanes vary from n-C₁₄ to n-C₂₀ (with very low abundance of alkanes of high M_r) maximising at n-C₁₇ or n-C₁₉.’. Therefore, we believe it is a useful and relevant reference in connection with algae and other ‘macrophytes’ (in the wider sense, at least) and directly connects with the discussion of the chemistry of algae in the main text.*

Line 258 and following: “C/N stable isotope analysis” is inaccurate, since C/N refers to the ratio of C to N. I suggest writing “C&N stable isotope ratio analysis” instead.

Reply: We agree with the reviewer here and have changed, as suggested. We thank the reviewer for this suggestion/correction.

Line 263 recent results from modern animals have shown that significant seaweed consumption can easily be identified in herbivores (sheep) using carbon isotope ratio analyses of bone collagen, bone apatite or tooth enamel. It is only in omnivorous (particularly human) diets, where seaweed constitutes a smaller extent of the diet, that identification of seaweed consumption is difficult. Isotopic variability between seaweeds of the same species are not such a large issue as painted in the manuscript: where isotopic variation is not systematic (so not e.g. only juveniles, or seaweeds at specific locations have lower isotope ratios), the mean is what affects the consumer’s isotope ratio, since people in the past could not selectively pick only those individuals of a species that have

comparatively high or low isotope ratios. Interindividual variation is thus not an issue if means are consistent. Additionally, significant isotopic variability between seaweed species is not that problematic, considering not all seaweeds are commonly consumed (and would not have been consumed in the past, e.g. chalky seaweeds are not particularly tasty or comfortable to eat, and some of the epiphytes are much too small and rare to have constituted a relevant part of past diets). Commonly consumed seaweeds like *Laminaria* spp, *Ulva* spp., *Palmaria palmata*, *Fucus* spp., *Himantalia elongata* etc. have generally elevated $\delta^{13}\text{C}$ values compared to terrestrial C3 plants (two of the few exceptions are *Porphyra* spp., and *Chondrus crispus* which are commonly eaten and have $\delta^{13}\text{C}$ values similar to terrestrial C3 plants). I still agree identifying seaweed consumption is difficult using stable isotope ratios though, because when diets contain less than ca. 20 % "marine" protein, this tends not to be identifiable due to isotopic variation in terrestrial plants, and also because it isn't possible to distinguish fish + terrestrial plants vs seaweed (other than *Porphyra* spp. and *Chondrus* c.) + meat using stable isotope ratios due to equifinality.

*Reply: We accept the points made above. Where seaweed constitutes a very significant part of an animal's diet, such as seaweed-eating sheep suggested here by the reviewer (Balasse et al 2005; Balasse et al 2009), stable isotopes can be reasonably expected to identify the consumption of seaweed. However, as the reviewer recognises, this is a herbivore where the discernment is between terrestrial C3 plants and seaweed. Again, as the reviewer acknowledges, our focus is human food consumption where an omnivorous diet makes the detection of seaweed difficult, if not impossible, if it's less than ~20% of the diet (still potentially notably significant from an anthropological point of view), as might reasonably be expected. We are suggesting the consumption of seaweed may have formed a significant part of the diet, but NOT that there were seaweed-eating humans in the same way Orkney has seaweed-eating sheep(!). It's clear there is some variability in isotopic signatures in both marine seaweeds and freshwater plants, as Raven et al 2002, cited in the text here and cited in the Balasse et al 2005 seaweed-eating sheep article, illustrates, although we have modified the main text from 'significant' to 'some', to address the concerns of the reviewer here that we might be over emphasising this point. It's also true that in general edible red seaweeds such as *Porphyra umbilicalis* and *Palmaria palmata* tend to have more negative values than green edible seaweeds, with brown in between. Although their values may be somewhat different from terrestrial C3 plants, within the context of an omnivorous diet where seaweed would be expected to be the main food consumed it's reasonable to suggest that isotopic analysis would not detect an anthropologically significant amount of seaweed in the diet, as the reviewer acknowledges above in their very helpful review. The red seaweeds we mention remain relevant, but we have amended the text on the isotopic data to balance the concerns of the reviewer and the valid points we feel are still relevant here.*

Line 274 "correlates with seaweed consumption" - Perhaps a better wording would be "consistent with seaweed consumption", since this is not a correlation in the statistical sense

Reply: We agree with the reviewer's suggestion here and have changed the wording to "is consistent with seaweed consumption", as advised.

Line 288 the term "depleted" is true for both high and low $\delta^{13}\text{C}$ values if it is not specified which isotope is depleted (12C or 13C). I'd suggest "can have notably lower $\delta^{13}\text{C}$ values"

Reply: We agree with the reviewer's suggestion here and have changed the wording to "can have notably lower $\delta^{13}\text{C}$ values", as advised.

Lines 290 and 291 “‰” signs missing

Reply: We have now corrected these omissions, as rightly pointed out.

Line 292 please add the word “ratio” before the brackets

Reply: We have now corrected this and thank the reviewer for pointing this omission out.

Line 297 the manuscript says “suggesting this practice [of freshwater plant consumption] certainly extended into the Middle Ages”. This implies continuity, but since this concerns three samples from Scotland and two from Lithuania, there is no continuity, since these places are so far apart, that they had little relation to each other. Whether or not there was an ebb and flow in freshwater plant consumption, or whether this was mostly continuous until it fell out of favour recently requires much further research.

Reply: We believe the reviewer makes a fair and valid point here and have reworded accordingly. What we have now emphasised is the continuity of the exploitation of aquatic resources in antiquity across Europe, despite their consumption as food being relatively rarely considered or recognised, which is an important and key point of this research, we believe. We also agree on the specific point that significant further research would be needed to gain a better and wider understanding of the degree of continuity in the consumption of seaweed and freshwater plants.

Line 299 I think it is worth quantifying your results here – 26 of 37 calculus samples suggest marine/freshwater plants/algae as human food. Also, considering the wording: only green seaweeds are technically plants

Reply: We have added the number of calculus samples with evidence for seaweed/algae freshwater plants, as suggested. This now has '21 of 37 calculus samples', since – as we discuss in the text – there is the possibility, at least, that the likely freshwater algae could have been consumed unintentionally from the bodies of freshwater these individuals drank from. As such, we would prefer to be cautious on this point and so only include those where intentional consumption can be reasonably inferred, which still results in a significant proportion of the calculus samples having evidence for marine macroalgae and freshwater plants as food. Our previous wording reflected the traditional view and usage of seaweeds as macrophytes/marine plants, but recognise current designations and nuances and have reworded accordingly. On this, green seaweeds are usually NOT regarded as plants (or animals), – although green seaweeds contains chlorophyll b (as well as the universal chlorophyll a for organisms which photosynthesise), like brown (a and c) and red (a and d) seaweeds many studies and scientists follow the strict designation of all seaweeds as algae, i.e. NOT plants, animals, fungi or bacteria. This being said, it is also useful to note that, even in more recent scientific studies, there remains NO convenient and singular consistent identification of seaweed (macroalgae), whether green or otherwise.

Line 307 RE use of wild resource to supplement agricultural produce - this was common everywhere in the Neolithic, considering domestic animals consumed mostly wild graze and browse

Reply: This is a reasonable comment, but there remain some senior and significant academics in this field who have been, and remain, slow to acknowledge this. We have added a comment in the final

paragraph to show that in fact supplementing agricultural produce with wild resources, continues today.

Line 316 Other strong arguments for seaweed consumption are that a) seaweed does not require irrigation, b) it does not require space on land to grow c) it does not require fertiliser. So sustainable seaweed farms end up being better for the environment than e.g. a wheat field.

Reply: We have added in 'sustainable' in our second last paragraph (new line number 349).

Line 319 this is not a grammatically complete sentence

Reply: We thank the reviewer for pointing this out – it was, in fact, a subtitle within the Methods section and our rewording and expansion of the Methods section has now addressed this issue.

Supplementary Information

The supplementary information has 199 pages, with 5333 lines. A simple “see also SI” or referencing “SI” in the manuscript is not that helpful, since finding the relevant section is difficult considering the sheer length of the SI. This has made reviewing this manuscript very arduous as well (in addition to its length). Please give page or paragraph numbers when referring to the supplementary information (if giving paragraph numbers, please include more numbered headings in the supplementary information). The supplementary information also often repeats itself and the manuscript word by word, please shorten it or put more headings in the table of contents and include hyperlinks. I would also prefer one long reference list at the end including all references rather than having several since this involves a lot of page turning to find the correct reference list.

Reply: We agree with the reviewer's comments here and will be more specific in our citation of the SI where that has not now been addressed by changes to the main text, as advised by reviewers and the editorial. We have also increased the number of headings in the table of contents, have more numbered headings in the text and have re-referenced the SI so the references are all now at the end of the SI, as sensibly advised.

Line 1297 and 1302 add “%”

Reply: We have now corrected these omissions, as rightly pointed out.

Line 4025 “it should be noted that if any marine component was fish or shellfish it would be expected that the isotopic data would reveal this.” Only if this consumption was larger than ca. 20 % of dietary protein when averaged over all seasons. Smaller amounts of marine food consumption are commonly not identifiable.

Reply: We accept this comment and we have amended the text accordingly, to acknowledge that only significant marine fish or shellfish consumption would be identified by the isotopic data. This would also be consistent with a small amount of marine fish or shellfish (the latter being easily collected in the intertidal zone, along with the seaweed) explaining the lipid profiles observed in the

Isbister sample DL5838 in Figure 3, yet not being discernible from isotopic data.

Line 4029 If large amounts of e.g. kelp were being consumed (let's say 50 % of the diet) throughout life, this would most certainly be visible in $\delta^{13}\text{C}$ values.

Reply: We agree with the comment here, but we were not suggesting kelp or another marine food constituting as much as half the diet being likely in the individuals/calculus samples in this study, with a more modest, yet still significant, two figure percentage more realistic.

See publications on modern seaweed-eating Orkney sheep. Seaweed consumption might be confused for fish consumption though. I fully agree with your disagreement in line 4032 with the paper's statement "...this population did not have any direct and regular dietary contact with the sea", since $\delta^{13}\text{C}$ values cannot be used to exclude marine food consumption since minor amounts would not be visible, and because some marine foods have $\delta^{13}\text{C}$ values similar to terrestrial C3 plants.

Reply: We have amended the text to make it clearer that marine consumption of up to 20%, which can be regarded as 'significant', would not be revealed by the isotopic data. This is NOT to dismiss the potential value of stable isotopic data, but highlight issues which can sometimes be overlooked, as the reviewer clearly explains, and this and earlier comments make it clear we agree with the reviewer on this point.

Line 4038 considering only one individual had equivocal evidence of marine food consumption, it might be a bit much to conclude the entire population at La Corona had dietary contact with the sea. A larger sample size would be needed for such a general assertion

Reply: We agree with the reviewer's comment here and have reworded the SI text to reflect this. Our main point was that this individual shows a dietary connection with the sea and therefore at least some of the population at La Corona may also have done so, despite the geographical location, although, as we say in the main text, it would have been in closer proximity to the sea in the Mesolithic period. However, as we now say, further research on more individuals would be necessary to test this, as the reviewer reasonably states.

Line 4382 I do not have access to this reference, but generally elemental data on bone is not reliable due to diagenesis and should be discounted (enamel would be ok for Sr, Ba and Ca). See also BURTON, J.H. & T.D. PRICE. 2006. The Use and Abuse of Trace Elements for Paleodietary Research, in Biogeochemical Approaches to Paleodietary Analysis: 159–71. https://doi.org/10.1007/0-306-47194-9_8

Reply: We felt the comparison with the elemental data was useful, given that its use with stable isotopic data is consistent with the clear chemical evidence for aquatic plants from the same archaeological site. However, we acknowledge what the reviewer says here and have removed that part of our original submission, as advised here. Consequently, we have removed this reference from the submission. The isotopic data from the Fontanals-Coll et al paper remains relevant and corroborates our own findings, with one individual (Skeleton No.27) from that study also being part of the research presented here, as we mention in the text, so we are able to make very similar points with this singular reference, without the inclusion of the potentially dubious reference discussing

elemental data on bone. We thank the reviewer for their advice here.

Line 4386 I think there is a typo here – what is ORSr?

Reply: ‘Observed ratio of strontium’ from the reference referred to in Line 4382, which has been used in connection with archaeological bone and palaeodietary reconstruction (not only in the study cited), but since we are happy to remove this part of the SI (and the associated reference), on the advice of the reviewer here, this is no longer relevant.

Line 4588 “extent

Reply: We have corrected this.

Reviewer #2 (Remarks to the Author):

The paper shows many analyses reporting chemical markers extracted from human dental calculus often related to compounds present in vegetable plants or animals. The results demonstrate that in Europe, in the Mesolithic - Early Middle Ages period, there was a widespread consumption of coastal resources (mainly seaweed and submerged aquatic plants). The paper may be accepted after minor revisions.

A general comment on biomarker. The authors report “..identifiable and characteristic biological marker (‘biomarker’) compounds extracted from...” It is not a biological marker but a chemical compound contained in a sample and, in this case, related to a food. The term biomarker as reported in the literature seems to be a fashion of the past! Anyway, biomarker can be accepted but please correctly define it.

Reply: We agree with the usefulness of defining ‘biomarker’ in this context, although this is a standard organic geochemical definition, also used widely in archaeological chemistry with the same meaning, and was used prior to the more recent use of ‘biomarker’ in the medical/proteomic sense, with many papers appearing in the journals ‘Nature’ and ‘Science’, for example, with the same definition of biomarker as is used here, primarily in an organic geochemical context. This was true particularly prior to the increasing use and publication of biomarkers in proteomics and DNA papers in the 1990s. Therefore, to that extent it could be regarded as ‘old fashioned’, but in the context of organic geochemistry, and the related archaeological chemistry, ‘biomarker’ retains its importance and significance and its meaning is clear in the context of studies using archaeological chemistry and drawing on organic compounds that are characteristic of the original molecule and survive over archaeological and geological time periods in a structurally recognisable form, allowing these ‘biomarkers’ to be correlated with the original source materials.

*In the context of the research presented here, and to reiterate, it requires a ‘biomarker’ approach, i.e. compounds characteristic of the original organic material **and** resistant to chemical and microbial degradation over archaeological and geological time periods. See, for example, Philp & Lewis (1987) ‘Organic Geochemistry of Biomarkers’ in Annual Reviews in Earth and Planetary Science, Vol .15, pp.363-395:*

‘A biomarker can be best thought of as an organic compound in a geological [or archaeological] sample that can be structurally related to its precursor molecule, which occurs as a natural product in a plant, animal, bacteria, spore, fungi, or any other potential source material. (The term biomarker is virtually synonymous with a number of other terms used in the literature, such as chemical fossil and biological marker, but for convenience biomarker is used throughout this article.)’

However, in recognition of the reviewer's comments here and more recent additional use of the word 'biomarker' we are happy to define biomarker in the text, in the context of this research, i.e. in the organic geochemical sense, still very much in use today. They are indeed also organic compounds, but the point remains that they are also chemically resilient and recognisable over archaeological time periods and beyond, hence 'biological marker compound', i.e. 'biomarker'.

Figure 2 at page 9. In the caption: Figure 13? Figure 2 is too much complex, I suggest to simplify reporting almost three reconstructed pyrograms with peak recognition

Reply: We believe the figure as it is best illustrates the multiple biomarker types, which collectively identify (red) seaweed in this sample (see also our 'point 2.' answer to Reviewer 1's comment concerning Lines 118 and 186). We are also mindful of the additional space it would require to CLEARLY display the three reconstructed pyrograms for Figure 2. However, if the additional space was to be available in the MAIN TEXT, without the removal of other figures, we could be persuaded of the merits of that, without any detrimental effect on other parts of the main text and its elements.

Page 12

Lines 153-156. Pyrograms report the presence of decomposition/rearrangement products of aminoacids: this discussion is misleading. DKPs are present both for animal and vegetable proteins and it is hardly understandable to assess their origin unless to perform aminoacid analysis or better proteomic.

*Reply: DKPs can indeed derive from both animal and vegetable proteins, as can other protein pyrolysis markers (see below), and, had there been sufficient sample sizes, amino acid analysis and/or proteomics would have been desirable, but the analyses carried out reflected the size of the calculus samples available **and** interest in a wide range of organic compounds, not only protein-derived material.*

However, the DKPs are a very minor part of the chemical compounds reported here and were not reported in the main text, so we wonder whether it is the DKPs specifically the reviewer has in mind, or the protein pyrolysis markers more generally? On the DKPs we, again, acknowledge there are, certainly in isolation, a number of possible origins, although in this study the predominance of the proline-glycine DKP, in addition to proline-alanine DKP at lesser abundance, but no other DKPs identified, suggests a collagen source (e.g. skin, bone), which is not particularly enlightening. While perhaps a small point, these DKPs occur in the TD-GC-MS analyses, rather than the Py-GC-MS, reflecting their formation at low – sub-pyrolysis – temperatures (e.g. see Buckley, Stott & Evershed, Analyst 1999). While the very narrow range of DKPs in this study limits their value, a wide range of DKPs can be related to the original amino acids constituting them, although NOT all amino acids produce observable DKPs in TD/Py-GC-MS analyses.

*The aromatic nitrogen-containing organic compounds are more significant in abundance and in the context of Py-GC-MS can be related to the aromatic amino acids constituting the original proteins, as we discuss and as numerous other studies using Py-GC-MS to analyse amino acids and proteins have shown (e.g. Kaal et al 2010; Simmonds 1970; Stankiewicz et al 1996 & 1997; etc. and cited in the text and SI). Consequently, we feel the reported findings are discussed appropriately, since it is the fact that the **connection** between notable abundances of aromatic nitrogen-containing organic compounds, which CAN be related to original amino acids, **and** the presence of the other biomarker groups, i.e. the acyl lipid-derived alkene/alkanes and the specific C₁-C₆ alkyl pyrroles, which provides*

a notable chemical correlation **collectively** and therefore justifies this discussion in the text; we agree that, in isolation, a discussion of the possible origin of the protein markers (including the DKPs) would not be particularly helpful.

Moreover, depending on pyrolysis temperature you may produce different molecules. Alkylpyrroles are often related to dairy products: how can you rule out their presence?

Reply: While pyrolysis temperature can have an impact on the molecules produced, this has been well studied and the reasons well understood. Some formations/transformations are more likely than others and this can be related to the chemistry of the organic materials analysed, so while a good chemical knowledge is essential, these analytical challenges, reasonably mentioned by the reviewer here, are not insurmountable and meaningful interpretations can be made if a biomarker approach (in the organic geochemical sense) and due caution is taken. Moreover, pyrolysis-GC-MS is used because in general terms pyrolysis of the biopolymers/organic macromolecules results in the carbon-carbon bonds of the polymeric materials being broken in a predictable way, albeit depending on the thermal energy, to produce the original building blocks constituting the original biopolymer/bound organic material, which, as sufficiently small molecules, can then be analysed by more conventional means such as GC-MS (used here) and so be directly related to the original material.

*Pyrrole itself and the methyl (C₁) and ethyl (C₂) pyrroles can have a number of possible origins, as we make clear in the SI. However, the far more unusual C₃ to C₆ alkylpyrroles (relating to the tetrapyrrole pigments in algae; see Sinninghe Damste et al 1992) can be related to a tetrapyrrole algal source. Dairy products would not produce these longer alkyl chain pyrroles, so while the far less specific C₁ and C₂ alkylpyrroles could, at least in part, derive from dairy this cannot be reliably determined and yet the C₃ to C₆ alkylpyrroles can be linked to a tetrapyrrole source such as seaweed or freshwater micro- or macroalgae. On the point the reviewer makes here, we actually state: 'Although C₁ and C₂ alkyl pyrroles can also derive from protein-derived amino acids such as glycine, proline, hydroxyproline, serine and glutamic acid^{12-14,20,40,41}, the C₃+ alkyl pyrroles identified in these six samples point to a tetrapyrrole source^{21,22,40,41}. (see SI). It is perhaps also worth noting that in the samples (across all sites) where a chemical signature for a freshwater plant (macrophyte) was observed there were no alkylpyrroles identified, so these organic heterocyclic compounds were by no means ubiquitous, even where other biomarkers/organic compounds survived. So the presence of the C₃ to C₆ alkylpyrroles excludes their derivation from a dairy source, though does not entirely exclude the possibility of dairy **in addition** to the seaweed/algae identified, but that would be speculative, which is why this possibility has not been discussed.*

Lines 160-163. Because fatty acids are not volatile, you cannot observe them in TD-GC-MS as well as in PY-GC-MS. Most probably they are not eluted, therefore you cannot sustain "...confirming an absence of these free lipid biomolecules".

Reply: This is incorrect. Fatty acids are sufficiently volatile to be thermally extracted and observed in the TD-GC-MS analyses, if present; indeed, FREE fatty acids were observed as the major free biomolecules in TD-GC-MS in previous studies (e.g. Buckley et al, Analyst 1999, Buckley & Evershed 2001). There is NO problem in the major fatty acids C_{16:0}, C_{18:1} and C_{18:0} fatty acids, for example, being volatilised at 310 deg C and eluting quantitatively, even if there is slight tailing of the peaks in the chromatogram due to their presence as the free acids (e.g. see Buckley et al 1999 for a good illustration of this, which also explains why they were purposefully not derivatised). Moreover, fatty acids can also be 'bound' within the polymeric organic component often found in archaeological samples and released ONLY at pyrolysis temperatures (~400+ deg C, typically ~600 deg C), so even

their origin as free fatty acids (at TD temperature) or 'bound' (which can include physically trapped, chemically bound as part of the biopolymeric component, or intact glycerides [though acyl lipids are often largely or fully hydrolysed when not physically absorbed by a protective mineral environment such as ceramic (clay) pottery]) can be discerned. Where we can agree with the reviewer here is when the fatty acids are appreciable polarised, such as dicarboxylic fatty acids, they do not elute, although hydroxy fatty acids formed via autoxidation can still be observed by TD-GC-MS, albeit not directly, as their dehydrated dienes derivatives (e.g. C_{18:1} 8,9,10 & 11-hydroxy fatty acids dehydrate to C_{18:2} fatty acids, eluding after the C_{18:0} fatty acid, often with the presence of several isomers in archaeological samples), formed in the pyroprobe at 310 deg C. What we are confirming is the absence of the major fatty acids, i.e. palmitic, oleic, stearic, etc., which normally DO dominate, hence we believe the wording remains appropriate here (even where the acyl lipids have been highly oxidised one would still expect a significant amount of palmitic acid if free biomolecules were present, which was patently not the case).

The sequence alkane/alkene of algal biopolymers should be demonstrated in your experimental conditions and compared with literature data: it is not clear if algae data reported in Figure 3 are obtained by the authors.

Reply: We agree this needs clarifying and have now addressed this. The data relates to the modelling, based on well established chemical pathways (see added references), of the predicted alkene/alkane distributions from green, brown and red seaweeds and compared with the alkene/alkane distributions of the calculus sample from Isbister 5838, which contains the highest abundance of organic material identified in the study. We have now linked this to relevant studies in both the text and the amended figure caption.

Supplementary data

The paper is well constructed, but too many results are discussed in the supplementary data with several repetitions to explain the presence of some "biomarkers". I suggest summarizing all the identified molecules per each sample and site in one table and avoid comments in the supplementary file.

Reply: We respectfully disagree with the reviewer on this point. This was to be comprehensive for each site, which may be of interest to archaeologists, in particular, who may have geographical areas of interest which are better served by the way we have presented it, ALTHOUGH we agree with the need to have an extended contents section, as advised by Reviewer 1, and putting the references in order as one list at the end of the SI. Some of the findings will rightly be regarded as being of insufficient interest for the main text, yet still useful for both context and interest to some. This also has the merit of showing the fuller picture – what isn't at one site may also be informing on the significance of what has been identified at other sites.

The figures are hardly understandable due to the many symbols. Please, simplify

Reply: Such complex chromatograms reflect the nature of the research and HAVE been published with this complexity previously. Simplifying them would necessarily mean losing relevant chemical evidence, making the value of the figure very limited, even if clearer at that point, so we would suggest maintaining the figures as they are, although their size should be considered so it can be maximised in the SI and therefore be more easily understood – some of the figures in the original submission have been smaller than the should/need to be.

CO₂ and acetonitrile could be the result of a laboratory pollution: have you daily checked the blanks?

*Reply: Not just daily, blanks were (and are) run between ALL samples and only when it is clear there will be NO carryover between samples is another sample analysed. There can be a little carryover after Py-GC-MS of a sample (depending on the abundance and chemical nature of the organic compounds present), but with two blanks run between samples the second blank is ALWAYS clean and there is **NO** carbon dioxide or acetonitrile in any of them. Moreover, the laboratory is used as a clean lab, specifically for this work, and so laboratory pollution is not a problem – acetonitrile is not used as a solvent (e.g. for HPLC) and there is no CO₂ gas in the lab, beyond levels in air. Indeed, the lack of even CO₂ in some samples does not support the suggestion by the reviewer here, although we do appreciate them raising the issue of potential contamination in a general sense, since it's something we take extremely seriously, as one might expect, and yet appears not to have been considered in similar studies by others on dental calculus.*

Reviewer #3 (Remarks to the Author):

This is an exciting paper, which offers evidence to critically evaluate the orthodoxy that there was a shift away from the use of marine resources in northwest Europe during the Neolithic. This orthodoxy is based on C/N isotopic analyses of skeletal populations, that can only offer broad brushstroke analyses of past diets and often underestimate the use of plants and algae. This paper instead uses TD-GC-MS and Py-GC-MS to investigate dental calculus from Mesolithic, Neolithic and Bronze Age and early Medieval skeletal remains, and demonstrates the ongoing consumption of seaweed into the Neolithic, as well as the use of freshwater plants into early Medieval times. The work is of significance to our understandings of the spread of agriculture in Europe. It is also significant methodologically, describing a pathway to better analyse the consumption of plants and algae in past diets. I am not a specialist in biomolecular archaeology and cannot comment on the merits of the chemistry used in this study. I can, however, comment on the research design and interpretation.

I have two minor comments:

1) At times the authors present an overview of an "agricultural revolution" that does not reflect the nuance now understood in global agricultural origins research. For example, "This represented a dramatic shift from all previous human existence which was based on hunting, gathering and non-accumulation, and established the social and economic foundations that underpin today's world, including the control and management of terrestrial food sources, land ownership, storable surplus, population increase and full sedentism," (lines 50-56). Whilst this was a significant point in human history, this sentence ignores the many "hunter-gatherer" economies that did store surpluses, and managed their landscapes and food resources, and the agropastoral economies that were not sedentary, etc. A little nuance here will go a long way.

2) There seems to be two stories presented in the paper, one about seaweed and another about freshwater aquatic plants. I believe these are not mutually exclusive stories and should be in the same paper. However, with the way parts of this is written the two claims seem to not be clearly demarcated, leading to some miscommunication. Line 299-301, "Three Chalcolithic/early Bronze age samples from inland sites in Scotland and two early Middle Age (5–6C AD) samples from Lithuania have biomarker evidence for submerged freshwater aquatic plant consumption, suggesting this

practice certainly extended into the Middle Ages." As this follows from a paragraph on the consumption of seaweed and is directly followed by, "Seaweed has been suggested as human food in antiquity before and consumption of marine resources is expected in the Mesolithic," it took me several reads to understand that "the practice extending into the Middle Ages" was not seaweed use. I suggest a small rewrite here and in the introductory paragraph where it feels like freshwater aquatic plant use is thrown in as an afterthought and it is confusing for the reader to follow and distinguish between the claims made.

Reply: We agree with the points made here and have amended accordingly.

Also, it should be Northwest Europe and Southwest Asia, not North West or Southwest.

Reply: We have now addressed this point.

Overall, this is an exciting paper that changes our picture of past diet and also has thoughtful forward focus.

** See Nature Portfolio's author and referees' website at www.nature.com/authors for information about policies, services and author benefits.

This email has been sent through the Springer Nature Tracking System NY-610A-NPG&MTS

Reviewers' Comments:

Reviewer #1:

Remarks to the Author:

I welcome the revisions made, they have addressed most of my concerns. I just have a few things left (line numbers refer to the new manuscript without tracked changes):

Apologies for not mentioning this in my first review, but I would really appreciate at least one (ideally more) example picture of calculus still on the tooth that would be suitable for analysis (it could go in the SI if there is no space in the main manuscript).

Re bilirubin: I think this should stay in the table, and am also happy about the additional explanation in the supplementary information.

Line 190: The sentence "When relative abundance of the sum of n-1-alkenes/n-alkanes for carbon chain numbers C8 to C16 is plotted for green, brown, red and sample DL5838, Isbister, Orkney" is missing the word seaweed.

Line 269 onward: There are still some inaccuracies in the discussion section on the use of stable isotope ratios (and the supplementary info line 3874) to detect seaweed consumption. As mentioned in my previous comments, isotopic variability is not the real issue for detecting most seaweed consumption (although it is for quantifying it), since most of the most popularly eaten seaweeds in Europe (e.g. *Laminaria* spp.) have higher carbon isotope ratios than most terrestrial foods (in C3 environments). There is isotopic variability in every species, this is not special to seaweeds. Elevated $\delta^{13}\text{C}$ values suggest seaweed consumption (or marine fish/mammal consumption, or C4...), regardless of isotope ratio variability. What is a problem is that bone collagen stable isotope ratios are determined by the stable isotope ratios of the dietary protein intake, and most dietary protein usually does not come from seaweeds, but from meat/fish. So in contrast to marine fish (also variable in isotope ratios!), seaweed is more easily overshadowed by other food sources. This is then combined with equifinality: Consuming marine fish and terrestrial plants can look the same as eating terrestrial plants, meat and seaweed (although you would need to eat a lot of seaweed). In principle I agree with your point that seaweed consumption by humans is difficult to identify by stable isotope ratios, but I think it is somewhat misleading as it is currently worded. Maybe just adding the issue of equifinality with respect to the consumption of other food sources would help? It would allow you to emphasise what's great about your method as well – stable isotope ratio analyses only give us one value per isotope ratio, but you get a whole suite of compounds to interpret, allowing a more nuanced picture. Just please don't misrepresent the issues in the use of stable isotope ratios in palaeodietary studies, it will only alienate a significant proportion of your readers, and confuse others.

Line 343: Re: use of wild resources: you only give mushrooms, shellfish and seaweed as examples, but actually, wild grasses are of much higher importance and should be mentioned here. E.g. alpine farmers almost exclusively make use of wild grasses in summers today.

Lines 280, 332, 333, 335: Ingestion of aquatic plants and seaweed: Technically your calculus work does not show evidence of ingestion (ingestion is defined as uptake into the body's digestive tract, involving swallowing), only of it being present near the teeth. I think it is reasonable to interpret that this means they were consumed, but it is an interpretation and I would separate these out (if you can, considering the word limit). I do realise this is nit-picking, but accuracy is a necessary part of science :)

Line 132: As pointed out by reviewer 2, the figure description for Figure 2 refers to Figure 13, when it should refer to Figure 2 (or just not refer to any figure, if it is the figure's caption anyway?).

There is a space missing between Figure and 2 in line 121, making "Figure 2" unsearchable in the

manuscript's main text (I also can't find a mention of Figure 3 in the main text, although it might also be somewhere with a typo).

Reviewer 2 asked about blanks, and this was replied to (satisfactorily I believe, though I'm no expert) in the response to reviewers, but no addition was made to the manuscript or the supplementary information as far as I can see. Since others will surely have the same concern, it seems worth adding this information on blanks/contamination issues to the manuscript/SI.

I agree with Reviewer 3's comments on adding some more nuance to the description of an "agricultural revolution". The authors reply that they have addressed this concern – but there appear to be no changes to the relevant sections – e.g. in the manuscript line 42 it still says "[A suite of domesticated crops and animals] gradually spread through Europe and was well established in southern Iberia by around 7500 years ago³³ and the far north of Scotland around 6000 years ago. This represented a dramatic shift from all previous human existence which was based on hunting, gathering and non-accumulation and established the social and economic foundations that underpin today's world, including the control and management of terrestrial food sources, land ownership, storable surplus, population increase and full sedentism." As Reviewer 3 points out, many "hunter-gatherer" economies that did store surpluses, and managed their landscapes and food resources, and there were agropastoral economies that were not sedentary, etc. This concern should be addressed.

Reviewer #2:

Remarks to the Author:

This interesting and exiting paper was improved and, practically the main observations have been discussed and changes have been done.

The added paragraph on analytical pyrolysis, useful for who does not know the technique, should cite a recent review on the instrumental methods together with n.40.

I understand that free unsaturated fatty acids (mainly palmitic and stearic acids) may be not detectable due to evaporation and extensive oxidation, but still for me it is hard to believe that they are completely absent. Being palmitic acid a common contaminant also in clean laboratories and released by bacteria, plant cells and so on, you should observe it (though in a low amount and as a broad peak) both in TD-GC-MS and Py-GC-MS. Probably, with an in-situ derivatization it should be detectable.

The paper can be published as it is.

Reviewer #3:

Remarks to the Author:

My comments have been addressed. However, see below for one minor, but important correction.

Abstract, Lines 9-12: This sentence does not make sense! At the moment it reads like you are saying that "A dramatic shift from the hunting, fishing and foraging that characterised all previous human existence" culminated in (i.e., resulted in) the Mesolithic! Which is obviously incorrect. Please rewrite.

REVIEWER COMMENTS

Reviewer #1 (Remarks to the Author):

I welcome the revisions made, they have addressed most of my concerns. I just have a few things left (line numbers refer to the new manuscript without tracked changes): We hope all our responses and explanations here address all remaining concerns and questions.

Apologies for not mentioning this in my first review, but I would really appreciate at least one (ideally more) example picture of calculus still on the tooth that would be suitable for analysis (it could go in the SI if there is no space in the main manuscript). We have added one new double figure showing two images of dental calculus from two different individuals. This will be Figure 1; a and b.

Re bilirubin: I think this should stay in the table, and am also happy about the additional explanation in the supplementary information. Thank you for your feedback here and we are pleased that the reviewer is happy to leave bilirubin in the table and that the additional explanation in the SI addresses any remaining concerns they may have had.

Line 190: The sentence “When relative abundance of the sum of n-1-alkenes/n-alkanes for carbon chain numbers C8 to C16 is plotted for green, brown, red and sample DL5838, Isbister, Orkney” is missing the word seaweed. We thank the reviewer for pointing this out and have now addressed this.

Line 269 onward: There are still some inaccuracies in the discussion section on the use of stable isotope ratios (and the supplementary info line 3874) to detect seaweed consumption. As mentioned in my previous comments, isotopic variability is not the real issue for detecting most seaweed consumption (although it is for quantifying it), since most of the most popularly eaten seaweeds in Europe (e.g. *Laminaria* spp.) have higher carbon isotope ratios than most terrestrial foods (in C3 environments). There is isotopic variability in every species, this is not special to seaweeds. We accept the reviewer’s general point here, but there is a particularly notable variation in aquatic freshwater plants and seaweeds, even within the same species, as studies have shown, with seasonality (in the exact same organism) and location being two recognised factors (e.g. Gichuki et al., *Hydrobiologia* 2001; Chappuis et al., *Freshwater Biology* 2017, in addition to refs 70-74 in the main text). These variations have been specifically used to suggest due caution when interpreting the significance of stable isotopic data for foodwebs and dietary/palaeodietary reconstruction, which is obviously of relevance here. **However**, we concede that any major consumption of seaweed would be observed in the stable isotopic data (although, as said previously, we would not reasonably expect a ~50% seaweed human diet, so our previous discussion was within the bounds of the credible and plausible) and we have now amended the main and SI text to reflect this. However, particularly given the more pronounced variation in freshwater plants (cf. seaweeds), which are an integral part of this study, we believe it is still appropriate to touch on the variability of carbon (and to a lesser extent nitrogen) isotope ratios in aquatic plants and (to a lesser degree) marine macroalgae, although we believe we have now put less emphasis on this. We have also included the reviewer’s point that if a food forms a moderate part of the diet (~20%) it is not likely to be detected by stable isotope analysis, which provides an overall and general picture of the diet, rather than what might be key foods consumed in moderate abundance.

Elevated $\delta^{13}\text{C}$ values suggest seaweed consumption (or marine fish/mammal consumption, or C4...), regardless of isotope ratio variability. What is a problem is that bone collagen stable isotope ratios are determined by the stable isotope ratios of the dietary protein intake, and most dietary protein usually does not come from seaweeds, but from meat/fish. So in contrast to marine fish (also variable in isotope ratios!), seaweed is more easily overshadowed by other food sources. This is then

combined with equifinality: Consuming marine fish and terrestrial plants can look the same as eating terrestrial plants, meat and seaweed (although you would need to eat a lot of seaweed). In principle I agree with your point that seaweed consumption by humans is difficult to identify by stable isotope ratios, but I think it is somewhat misleading as it is currently worded. We agree with the reviewer's point here and have amended the text accordingly.

Maybe just adding the issue of equifinality with respect to the consumption of other food sources would help? It would allow you to emphasise what's great about your method as well – stable isotope ratio analyses only give us one value per isotope ratio, but you get a whole suite of compounds to interpret, allowing a more nuanced picture. Just please don't misrepresent the issues in the use of stable isotope ratios in palaeodietary studies, it will only alienate a significant proportion of your readers, and confuse others. We believe the reviewer's comments here are fair and have amended the text accordingly and therefore hope all previous concerns on these points have now been addressed.

Line 343: Re: use of wild resources: you only give mushrooms, shellfish and seaweed as examples, but actually, wild grasses are of much higher importance and should be mentioned here. E.g. alpine farmers almost exclusively make use of wild grasses in summers today. We agree that wild grasses are still used; however, the very widespread collection and consumption of wild mushrooms across much of central Europe and in Mediterranean countries, and the exploitation of wild coastal resources, particularly fish and shellfish, along most of the European Atlantic coast, is also very extensive. We have added in 'wild grasses' to the list of currently used wild resources.

Lines 280, 332, 333, 335: Ingestion of aquatic plants and seaweed: Technically your calculus work does not show evidence of ingestion (ingestion is defined as uptake into the body's digestive tract, involving swallowing), only of it being present near the teeth. I think it is reasonable to interpret that this means they were consumed, but it is an interpretation and I would separate these out (if you can, considering the word limit). I do realise this is nit-picking, but accuracy is a necessary part of science :) The point made here is understood; it is, of course, an interpretation that edible items that entered the mouth were ultimately swallowed. However, in studies of the contents of archaeological pottery/ceramics, these are widely considered as human food, and published as such, rather than for domestic animals, or prepared for ritualistic reasons, etc. So while the reviewer makes a good and pertinent point here, which we acknowledge, we feel that since residues found in pottery vessels are widely accepted as representing aspects of human diet, edible items found embedded in an individual's dental calculus, can be considered as very likely to have been ingested. We have amended the text in one place to make it clear that items that edible items that entered the mouth were *most probably* ingested.

Line 132: As pointed out by reviewer 2, the figure description for Figure 2 refers to Figure 13, when it should refer to Figure 2 (or just not refer to any figure, if it is the figure's caption anyway?). We have amended/corrected this and thank the reviewer for pointing this out.

There is a space missing between Figure and 2 in line 121, making "Figure 2" unsearchable in the manuscript's main text (I also can't find a mention of Figure 3 in the main text, although it might also be somewhere with a typo). We have amended/corrected this and thank the reviewer for pointing this out.

Reviewer 2 asked about blanks, and this was replied to (satisfactorily I believe, though I'm no expert) in the response to reviewers, but no addition was made to the manuscript or the supplementary information as far as I can see. Since others will surely have the same concern, it seems worth adding this information on blanks/contamination issues to the manuscript/Sl. We agree with the

point made here and have now added the use of blanks in this study to the manuscript and explained their significance for excluding the possibility of contamination. We believe this is a helpful addition and so are happy to address the concerns of both reviewers in this respect, which we agree will be helpful for the readers interested in this research.

I agree with Reviewer 3's comments on adding some more nuance to the description of an "agricultural revolution". The authors reply that they have addressed this concern – but there appear to be no changes to the relevant sections – e.g. in the manuscript line 42 it still says "[A suite of domesticated crops and animals] gradually spread through Europe and was well established in southern Iberia by around 7500 years ago³³ and the far north of Scotland around 6000 years ago. This represented a dramatic shift from all previous human existence which was based on hunting, gathering and non-accumulation and established the social and economic foundations that underpin today's world, including the control and management of terrestrial food sources, land ownership, storable surplus, population increase and full sedentism." As Reviewer 3 points out, many "hunter-gatherer" economies that did store surpluses, and managed their landscapes and food resources, and there were agropastoral economies that were not sedentary, etc. This concern should be addressed. – We have modified the text to highlight that it is the combination of different features, some of which could have been present separately beforehand, that created the shift that ultimately led to the change in fundamental social and economic conditions.

Reviewer #2 (Remarks to the Author):

This interesting and exiting paper was improved and, practically the main observations have been discussed and changes have been done. - We thank the reviewer for their positive comments here. The added paragraph on analytical pyrolysis, useful for who does not know the technique, should cite a recent review on the instrumental methods together with n.40. – NO review exists, as such. This reflects the diverse applications of analytical pyrolysis (including thermal desorption, whether used sequentially with pyrolysis-GC-MS or not) and the diverse nature of the materials analysed, meaning an overall review in its strictest sense doesn't exist. Reviews have been on specific applications (e.g. microplastics) or materials (e.g. lacquers). Neither of these examples (or others) is useful as the review [reasonably] suggested by the reviewer here. Analytical applications include microbiology, microplastics, environmental contamination/applications, petroleum exploration/organic geochemistry, forensics (e.g. paints, cosmetics) and others, but each exists within its own area and any reviews reflect this. The closest area, perhaps, is the use of Py-GC-MS (although without sequential TD-GC-MS, so no thermal extraction of any free biomolecules prior to pyrolysis) in the study of cultural materials, e.g. paintings, ambers, lacquers, but not commonly to date to archaeological materials with a complex and relatively diverse organic chemistry. Consequently, no meaningful review is available in a form one might wish, certainly on the use of sequential TD-GC-MS and Py-GC-MS, as is employed in this study. However, noting the reasonable suggestion here we have now added a paper using thermal desorption combined with pyrolysis-GC-MS to determine free organic compounds and the nature of polymeric material present in plastics (this is NOT archaeological, BUT does, relatively unusually, combine TD- and Py- with GC-MS) AND have now cited what is considered THE **Bible** on analytical pyrolysis – the Analytical Pyrolysis Handbook (3rd Edition), which was published in 2021. This provides the latest and most comprehensive 'review' and overview of the technique of analytical pyrolysis and, combined with the existing reference, with its archaeological application, and the Molecules 2020 paper now included, provides a fairly comprehensive source of information for those with an interest in the instrumental methods and technique(s). We have consulted with THE world-leading experts in this field, i.e. Analytix and CDS, and they have confirmed there is no current review of analytical pyrolysis as a whole, beyond the Analytical Pyrolysis Hanbook, now cited in the manuscript. We agree with

the usefulness of a 'review' of the analytical approach in some sense, so hope these additions now address this point, given the realities as they are.

I understand that free unsaturated fatty acids (mainly palmitic and stearic acids) may be not detectable due to evaporation and extensive oxidation, but still for me it is hard to believe that they are completely absent. Being palmitic acid a common contaminant also in clean laboratories and released by bacteria, plant cells and so on, you should observe it (though in a low amount and as a broad peak) both in TD-GC-MS and Py-GC-MS. Probably, with an in-situ derivatization it should be detectable. NO, while we understand the reviewer's points here this is not entirely correct. Much of the problem here is that the analysis of dental calculus using this relatively niche technique remains poorly understood by many. It is true that palmitic and stearic acids CAN be common contaminants in a laboratory environment. However, in a clean lab, as has been used in this case, the risk of contamination can be minimised and laboratory blanks are routinely run to check this. On a pedantic point palmitic and stearic acids are NOT 'free unsaturated acids'; they are 'free saturated acids', which we're sure is what the reviewer meant. The 'free unsaturated fatty acids' can oxidise relatively easily, but the free saturated fatty acids, palmitic and stearic, are very stable and would not be expected to easily evaporate or oxidise under normal conditions, including many archaeological contexts. To this extent we appreciate the reviewer's comment, although it is possible to exclude exogenous sources of these common fatty acids if they are not an inherent part of the analytical procedure (e.g. some solid phase extraction columns can have a disturbingly high amount of palmitic and stearic acids!), so we are acutely aware of this potential problem. Moreover, the palmitic acid would not be an especially broad peak in either the TD-GC-MS or Py-GC-MS, as previous studies have shown (e.g. Buckley et al., Analyst 1999) – there is some tailing, but palmitic acid elutes from the GC column and if present in trace amount could be detected using mass chromatograms. In the context of calculus in-situ derivatisation would NOT allow the detection of palmitic acid unless it was close to the limit of detection and as the 1999 paper illustrated, underivatised free fatty acids are easily observable where present, albeit with 'ugly' tailing peaks from a chromatographic perspective. Moreover, palmitic acid is often absent from the Py-GC-MS, while observed in the TD-GC-MS, although it can indeed be observable in both, where present, and this is relatively commonly observed where there has been NO derivatisation, reflecting the fact that palmitic (and often stearic) are commonly major constituents in many archaeological organic residues analysed this way, as the reviewer clearly recognises. IMPORTANTLY, in many northern European and Iberians sites palmitic and stearic acids are entirely absent (undetectable) in calculus samples. HOWEVER, in eastern Mediterranean sites palmitic and stearic acids, or [often] their amide or nitrile analogues, ARE the major compounds present in the TD-GC-MS and often (though less so) significant in the Py-GC-MS – this despite NO in-situ derivatisation. This points to the presence of water/moisture in the northern European and Iberian calculus samples allowing bacteria to degrade the organic component to the point at which there are essentially no free biomolecules, yet the polymeric component remains in many of these samples because it is intractable as far as bacteria are concerned, as has been noted in organic geochemical contexts by researchers studying biomolecules, including biopolymers, using very similar techniques. In contrast, calculus from eastern Mediterranean sites contain an abundance of both free and biopolymeric/bound organic material, reflecting the greater aridity at these sites and so preventing microbial degradation of the free biomolecules, including palmitic and stearic acids and their derivatives (e.g. their amides and nitriles). Clearly the issues raised here would be of interest to many scientific researchers and archaeologists and it is hoped a paper will result from this ongoing research, although it is obviously beyond the remit of this study, but having analysed over 300 samples from over 50 sites it's clear that preservation of the organic component is highly site-dependent, particularly in relation to free organic compounds present. However, despite much of what we say here we DO appreciate the reviewer's concerns and we welcome the scrutiny in ensuring these sort of potential issues have been addressed.

The paper can be published as it is.

Reviewer #3 (Remarks to the Author):

My comments have been addressed. However, see below for one minor, but important correction.

Abstract, Lines 9-12: This sentence does not make sense! At the moment it reads like you are saying that "A dramatic shift from the hunting, fishing and foraging that characterised all previous human existence" culminated in (i.e., resulted in) the Mesolithic! Which is obviously incorrect. Please rewrite. Thank you for highlighting this, we have changed and corrected it.

** See Nature Portfolio's author and referees' website at www.nature.com/authors for information about policies, services and author benefits.

This email has been sent through the Springer Nature Tracking System NY-610A-NPG&MTS

Reviewers' Comments:

Reviewer #1:

Remarks to the Author:

All in all I am satisfied with the changes, and the paper can be published as is.

Response to Reviewers

Reviewer #1 (Remarks to the Author):

All in all I am satisfied with the changes, and the paper can be published as is.

We are proceeding accordingly.